# WHEN NARROWER IS BETTER: THE NARROW WIDTH LIMIT OF BAYESIAN PARALLEL BRANCHING NEURAL NETWORKS

**Zechen Zhang**
Department of Physics, Harvard University
Center for Brain Science, Harvard University
zechen_zhang@g.harvard.edu

**Haim Sompolinsky**
Center for Brain Science, Harvard University
Kempner Institute
Edmond and Lily Safra Center for Brain Sciences,
Hebrew University of Jerusalem
hsompolinsky@mcb.harvard.edu

## ABSTRACT

The infinite width limit of random neural networks is known to result in Neural Networks as Gaussian Process (NNGP) (Lee et al. (2018)), characterized by task-independent kernels. It is widely accepted that larger network widths contribute to improved generalization (Park et al. (2019)). However, this work challenges this notion by investigating the narrow width limit of the Bayesian Parallel Branching Neural Network (BPB-NN), an architecture that resembles neural networks with residual blocks. We demonstrate that when the width of a BPB-NN is significantly smaller compared to the number of training examples, each branch exhibits more robust learning due to a symmetry breaking of branches in kernel renormalization. Surprisingly, the performance of a BPB-NN in the narrow width limit is generally superior to or comparable to that achieved in the wide width limit in bias-limited scenarios. Furthermore, the readout norms of each branch in the narrow width limit are mostly independent of the architectural hyperparameters but generally reflective of the nature of the data. We demonstrate such phenomenon primarily for branching graph neural networks, where each branch represents a different order of convolutions of the graph; we also extend the results to other more general architectures such as the residual-MLP and show that the narrow width effect is a general feature of the branching networks. Our results characterize a newly defined narrow-width regime for parallel branching networks in general.

## 1 INTRODUCTION

The study of neural network architectures has seen substantial growth, particularly in understanding how network width impacts learning and generalization. In general, wider networks are believed to perform better (Allen-Zhu et al. (2019); Jacot et al. (2018); Gao et al. (2024)). However, this work challenges the prevailing assumption by exploring the narrow width limit of Bayesian Parallel Branching Neural Network (BPB-NN), an architecture inspired by neural networks with residual blocks (He et al. (2016); Chen et al. (2020a; 2022)). We show theoretically and empirically that narrow-width BPB-NNs can perform better than their wider counterparts due to the symmetry-breaking effect in kernel renormalization, in bias-limited scenarios. We present a detailed analysis of BPB-NNs primarily by exploring the Bayesian Parallel Branching Graph Convolution Networks (BPB-GCN), while extending the results to other more general architectures such as the residual-MLP in the appendix.

**Contributions** :

1. We introduce a novel yet simple graph neural architecture with parallel independent branches, and derive the exact generalization error for node regression in the statistical limit as the sample size $P \to \infty$ and network width $N \to \infty$, with their ratio a finite number $\alpha = P/N$, in the over-parameterized regime.

2. We show that in the Bayesian setting the bias will decrease and saturate at a narrow hidden layer width, a surprising phenomenon due to kernel renormalization. We demonstrate that this can be understood as a robust learning effect of each branch in the student-teacher task, where each student's branch is learning the teacher's branch.

3. We demonstrate this narrow-width limit in the real-world dataset Cora and understand each branch's importance as a nature of the dataset.

4. We further show that this narrow-width effect is a general feature of Bayesian parallel branching neural networks (BPB-NNs), with the residual-MLP architecture as an example.

## 2 RELATED WORKS

**Infinitely wide neural networks:**  Our work follows a long tradition of mathematical analysis of infinitely-wide neural networks (Neal (2012); Jacot et al. (2018); Lee et al. (2018); Bahri et al. (2024)), resulting in NTK or NNGP kernels. Recently, such analysis has been extended to structured neural networks, including GCNs (Du et al. (2019); Walker & Glocker (2019); Huang et al. (2021)). However, they do not provide an analysis of feature learning in which the kernel depends on the tasks.

**Kernel renormalization and feature learning:**  There has been progress in understanding simple MLPs in the feature-learning regime as the shape of the kernel changes with task or time (Li & Sompolinsky (2021); Atanasov et al. (2021); Avidan et al. (2023); Wang & Jacot (2023)).  We develop such understanding in graph-based networks.

**Theoretical analysis of GCN:**  There is a long line of works that theoretically analyze the expressiveness (Xu et al. (2018); Geerts & Reutter (2022)) and generalization performance (Tang & Liu (2023); Garg et al. (2020); Aminian et al. (2024)) of GCN. However, it is challenging to calculate the dependence of generalization errors on tasks. In particular, the PAC-Bayes approach Liao et al. (2020); Ju et al. (2023) results in generalization bounds that are too large and that can be only computed with norms of learned weights. To our knowledge, our work is first to decompose the generalization error into bias and variance *a priori* (not dependent on learned weights) for linear GCNs with residual-like structures. The architecture closest to our linear BPB-GCN is the linearly decoupled GCN proposed by Cong et al. (2021); however, the overall readout vector is shared for all branches, which will not result in kernel renormalization for different branches.

## 3 BPB-GCN

### 3.1 PARALLEL BRANCHING GCN ARCHITECTURE

We are motivated to study the parallel branching networks as they resemble residual blocks in commonly used architectures (Kipf & Welling (2016); He et al. (2016); Chen et al. (2020b)) and are easy to study analytically with our Bayesian framework. We primarily focus on the graph setting (BPB-GCN) in the main sections, and discuss the more general case in appendix B.

Given graph $G = (A, X)$, where $A$ is the adjacency matrix and $X$ the node feature matrix, the final readout for node $\mu$ is a scalar $f^\mu(G; \Theta)$ that depends on the graph and network parameters $\Theta$. The parallel branching GCN is an ensemble of GCN branches, where each branch operates independently with no weight sharing. In this work, we focus on the simple setup of branches made of linear GCN with one hidden layer, but with different number of convolutions $A^1$ on the input node features (Figure 1(a)). In this way, parallel branching GCN is analogous to GCN with residual connections, for which the final node readout can also be thought of as an ensemble of convolution layers (Veit et al. (2016)). Concretely, the overall readout $f^\mu(G; \Theta)$ for node $\mu$ is a sum of $L$ branch

---

[1]In this paper, the convolution operation is normalized as $A = D^{-1/2}(\hat{A} + I)D^{-1/2}$, where $\hat{A}$ is the original Adjacency matrix and $D$ is the degree matrix. We also use feature standardization after convolution for each branch to normalize the input

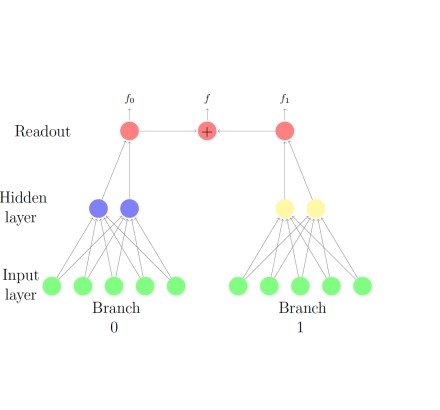 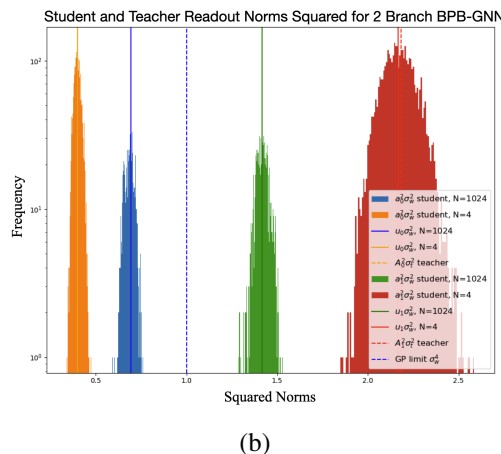

(a)                      (b)

Figure 1: Overview of the main takeaway: BPB-GCN learns robust representations for each branch at narrow width. (a) The parallel branching GCN architecture, with 2 branches. The independent branches have non-sharing weights and produce the final output $f$ as a sum of branch-level readouts $f_l$. (b) Student and teacher readout norms squared for wide and narrow student BPB-GCN networks. The student network with width $N$ is trained with the teacher network's output. Histograms correspond to the samples from Hamiltonian Monte Carlo simulations and solid lines correspond to the order parameters calculated theoretically. $\sigma_t = \sigma_w = 1$. At $N = 4$, the HMC samples of branch readout norms squared (orange and red histograms) for the student network $\frac{\|a_l\|^2}{N}\sigma_w^2$ concentrate at their respective theoretical values $u_l\sigma_w^2$ and overlap with the teacher's readout norms squared $\frac{\|A_l\|^2}{N}\sigma_t^2$ (orange and red dashed lines) for corresponding branches. At $N = 1024$ the samples for the student network (blue and green histograms) concentrate at their respective theoretical values but remain far from the teacher's values, instead approaching the GP limit $\sigma_w^4$ (blue dashed line).

readouts $f_l^\mu$:

$$f^\mu(G; \Theta) = \sum_{l=0}^{L-1} f_l^\mu(G; \Theta_l = \{W^{(l)}, a^{(l)}\}), \tag{1}$$

where

$$f_l^\mu(G, \Theta_l) = \frac{1}{\sqrt{L}} \sum_{i=1}^{N} \frac{1}{\sqrt{N}} a_i^{(l)} \sum_{j=1}^{N_0} \frac{1}{\sqrt{N_0}} W_{ij}^{(l)} \sum_{\nu=1}^{n} (A^l)_{\mu\nu} x_j^\nu \tag{2}$$

In matrix notation,

$$F = \sum_l \frac{1}{\sqrt{LNN_0}} A^l X W^{(l)} a^{(l)} \tag{3}$$

Note that when $L = 2$, the network reduces exactly to a 2-layer residual GCN (Chen et al. (2020c)). Here $N_0$ is the input dimension, $N$ the width of the hidden layer, $W^{(l)}$ the weight of the hidden layer for branch $l$, $A^l$ the $l$th power of the adjacency matrix and $a^{(l)}$ the final readout weight for branch $l$. We will consider only the linear activation function for this paper and provide a summary of notations in Appendix A.1.

## 3.2 BAYESIAN NODE REGRESSION

We consider a Bayesian semi-supervised node regression problem, for which the posterior probability distribution for the weight parameters is given by

$$P(\Theta) = \frac{1}{Z} e^{-E(\Theta; G, Y)/T} = \frac{1}{Z} \exp\left(-\frac{1}{2T} \sum_{\mu=1}^{P} (f^\mu(G, \Theta) - y^\mu)^2 - \frac{1}{2\sigma_w^2} \Theta^T \Theta\right), \tag{4}$$

where the first term in the exponent corresponds to the likelihood term induced by learning $P$ node labels $y_\mu$ with squared loss and the second term corresponds to the Gaussian prior with variance

$\sigma_w^2$. $Z = \int e^{-E(\Theta)/T} d\Theta$ is the normalization constant. This Bayesian setup is well motivated, as the Langevin dynamics trained with energy potential $E$ and temperature $T$ that results in this equilibrium posterior distribution shares a lot in common with the Gradient Descent (Avidan et al. (2023); Naveh et al. (2021)) and Stochastic GD optimizers (Mignacco & Urbani (2022); Wu et al. (2020)). In fact, Li & Sompolinsky (2021) shows empirically that the Bayesian equilibrium is a statistical distribution of the usual gradient descent with early stopping optimization and with random initializations at 0 temperature in DNNs, where the $L_2$ regularization strength $\sigma_w^2$ corresponds to the Gaussian initialization variance.

We are interested in understanding the weight and predictor statistics of each branch and how they contribute to the overall generalization performance of the network. In the following theoretical derivations, our working regime is in the overparameterizing high-dimensional limit (Li & Sompolinsky (2021); Montanari & Subag (2023); Bordelon & Pehlevan (2022); Howard et al. (2024)): $P, N, N_0 \to \infty$, $\frac{P}{N} = \alpha$ finite and the capacity $\alpha_0 = \frac{P}{LN_0} < 1$. As we will show later, this limit is practically true even with $P, N$ not so large (our smallest $N$ is 4). We will also use near-0 temperature, in which case the training error will be near 0.

## 3.3 KERNEL RENORMALIZATION AND ORDER PARAMETERS

The normalization factor, or the partition function, $Z = \int e^{-E(\Theta)/T} d\Theta$ carries all the information to calculate the predictor statistics and the generalization dependence on network hyperparameters $N, L, \sigma_w^2$. Using Eq. 2,4, we can integrate out the readout weights $a_l$'s first, resulting in

$$Z = \int dW e^{-H(W)}, \tag{5}$$

with effective Hamiltonian $H(W)$ in terms of the hidden layer weights for all branches

$$H(W) = \frac{1}{2\sigma_w^2} \sum_{l=0}^{L-1} \text{Tr} W_l^T W_l + \frac{1}{2} Y^T (K(W) + TI)^{-1} Y + \frac{1}{2} \log \det(K(W) + TI), \tag{6}$$

where

$$K(W) = \frac{1}{L} \sum_l \frac{\sigma_w^2}{N} (H_l(W_l) H_l(W_l)^T)|_P \tag{7}$$

is the $P \times P$ kernel matrix dependent on the observed $P$ nodes with node features $H_l = A^l X W_l$ and we denote $|_P$ as the matrix restricting to the elements generated by the training nodes.

As shown in Appendix A.2, we can further integrate out the $W_l$'s and get the partition function $Z = \int Du e^{-H(u)}$ described by a final effective Hamiltonian independent of weights

$$H(u) = S(u) + E(u), \tag{8}$$

where we call $S(u)$ the entropic term

$$S(u) = -\sum_l \frac{N}{2} \log u_l + \sum_l \frac{N}{2\sigma_w^2} u_l \tag{9}$$

and $E(u)$ the energetic term

$$E(u) = \frac{N\alpha}{2P} Y^T (\sum_l \frac{1}{L} u_l K_l + TI)^{-1} Y + \frac{N\alpha}{2P} \log \det(\sum_l \frac{1}{L} u_l K_l + TI), \tag{10}$$

where $K_l = \frac{\sigma^2}{N_0} [A^l X X^T A^l]|_P$ is the $(P \times P)$ input node feature kernel for branch $l$.

Therefore, the final effective Hamiltonian has the overall kernel

$$K = \sum_l \frac{1}{L} u_l K_l, \tag{11}$$

where $u_l$'s are order parameters which are the minimum of the effective Hamiltonian Eq. 8 that satisfy the saddle point equations:

$$N(1 - \frac{u_l}{\sigma_w^2}) = -r_l + \text{Tr}_l, \tag{12}$$

where $r_l = Y^T (K + TI)^{-1} \frac{u_l K_l}{L} (K + TI)^{-1} Y$ and $\text{Tr}_l = \text{Tr}[K^{-1} \frac{u_l K_l}{L}]$.

**GP kernel vs. renormalized kernel:** Observe that as $\alpha = P/N \to 0$, the entropic term dominates, and thus $u_l = \sigma_w^2$ for all branches from Eq. 12. This is the usual Gaussian Process (GP) limit, or the infinite-width limit. In this case, the kernel $K_l$ for each branch is not changed by the training data, and each branch has the same contribution in terms of its strength.

However, when $\alpha = P/N$ is large, there is a correction to the GP prediction, and $u_l$'s in general depend on the training data; therefore, we have feature learning in each branch with kernel renormalization. It turns out that $u_l$ is exactly the statistical average of readout norm squared over the posterior distribution (Appendix A.4), ie.

$$u_l = \langle \|a_l\|^2 \rangle / N \tag{13}$$

for each branch $l$, where we use the bracket notation to represent the expectation value over the posterior distribution. Therefore, branches in general become more and more different as the width of the hidden layer $N$ decreases. We call this phenomenon *symmetry breaking*, which is discussed in Section 4.2.

### 3.4 PREDICTOR STATISTICS AND GENERALIZATION

Under the theoretical framework, we obtain analytically (Appendix A.3) the mean predictor $\langle f^\nu(G) \rangle$ and variance $\langle \delta f_\nu(G)^2 \rangle$ for a single test node $\nu$ as Eq.54 and Eq.55, respectively. We use this to calculate the generalization performance, defined by the MSE on $t$ test nodes

$$\epsilon_g = \langle \frac{1}{t} \sum_{\nu=1}^{t} (f^\nu(G) - y^\nu)^2 \rangle_\Theta = \text{Bias} + \text{Variance}, \tag{14}$$

where

$$\text{Bias} = \frac{1}{t} \sum_{\nu=1}^{t} (\langle f^\nu(G) \rangle_\Theta - y^\nu)^2, \text{Variance} = \frac{1}{t} \sum_{\nu=1}^{t} \langle \delta f_\nu^2 \rangle. \tag{15}$$

Note that our definition of bias and variance is a statistical average over the posterior weight distribution, which is slightly different from the usual definition from GD.

## 4 THE NARROW WIDTH LIMIT

As we discussed briefly in Section 3.3, the kernel becomes highly renormalized at narrow width. In fact, in the extreme scenario as $\alpha = P/N \to \infty$, the energetic term in the Hamiltonian completely dominates, and we would expect that the generalization performance saturates as the order parameters in the energetic terms become independent of width $N$. Therefore, just as infinitely wide networks correspond to the GP limit, we propose that there exists a ***narrow width limit*** when the network width is extremely small compared to the number of training samples.

### 4.1 ROBUST LEARNING OF BRANCHES: THE EQUIPARTITION CONJECTURE

What happens in the narrow width limit? In the following, we demonstrate that each branch will learn robustly at narrow width.

**The equipartition theorem** Consider a student-teacher network setup, where the teacher network is given by

$$f^*(G; \Theta^*) = \sum_l f_l^*(G; W_l^*) = \sum_l \frac{1}{\sqrt{N_t L}} \sum_i a_{i,l}^* h_i^l(G; W_l^*). \tag{16}$$

$W_{ij,l}^* \sim \mathcal{N}(0, \sigma_t^2)$ and $a_{i,l}^* \sim \mathcal{N}(0, \beta_l^2)$, where $\beta_l^2$ is the variance assigned to the readout weight for the teacher branch $l$ and $N_t$ is the width of the hidden layer. Similarly, the student network is given by the same architecture, with layer width $N$ and learns from $P$ node labels from the teacher $Y_\mu = f_\mu^*(G, \Theta^*)$ in the Bayesian regression setup of Eq. 4 with prior variance $\sigma_w^2$. We state that at $T = 0$ temperature, in the limit $N_t \to \infty$ and $\alpha = P/N \to \infty$, there exists a solution to the saddle point equations 12

$$u_l = \sigma_t^2 \beta_l^2 / \sigma_w^2. \tag{17}$$

*Proof*:

At the narrow width limit $\alpha \to \infty$, the saddle equation 12 becomes $r_l = \text{Tr}_l$. Now we calculate

$$Y_\mu^* Y_\mu^{*T} = \sum_{i,j=1}^{N_t} \sum_{l_1,l_2} \frac{1}{N_t L} a_{i,l_1}^* a_{j,l_2}^* h_i^{l_1,\mu}(G; W_l) h_j^{l_2,\nu}(G; W_l) \tag{18}$$

by law of large numbers, this quantity concentrates at its expectation value

$$Y^* Y^{*T} = \lim_{N_t \to \infty} \mathbb{E}_{a^*,W^*}(YY^T) = \sum_l \frac{\beta_l^2 \sigma_t^2}{N_0 L} (A^l X X^T A^l)|_P = \frac{\beta_l^2}{L} K_l(\sigma_t^2), \tag{19}$$

where $K_l(\sigma^2)$ represents the GP input kernel with prior variance $\sigma^2$. Therefore, $r_l$ becomes

$$r_l = Y^{*T} K^{-1} \frac{u_l K_l}{L} K^{-1} Y^* = \text{Tr}(K^{-1} \frac{u_l K_l}{L} Y^* Y^{*T}) = \text{Tr}_l((\sum_l u_l K_l(\sigma_w^2)/L)^{-1} \sum_l \beta_l^2 \frac{K_l}{L}) \tag{20}$$

Thus there exists a solution that satisfies the saddle point euations

$$u_l \sigma_w^2 = \beta_l^2 \sigma_t^2. \tag{21}$$

Furthermore, by Eq.13, we show that the student branch norms learn exactly the teacher branch norms, ie. $\langle a_l^2 \rangle \sigma_w^2 = a_l^{*2} \beta_l^2$ . We call this equipartition theorem, as the mean-squared readout and the variance (A.4) have to exactly balance each other, which contribute to the energy term in the Hamiltonian. Although the theorem is proven for the BPB-GCN architecture, we show that the result is general in Appendix B.1.

## 4.2 STUDENT-TEACHER EXPERIMENT ON ROBUST BRANCH LEARNING

We demonstrate this robust learning phenomenon and provide a first evidence of the equipartition conjecture with the student-teacher experiment setup introduced in the previous section. We use the contextual stochastic block model (CSBM) (Deshpande et al. (2018)) to generate the graph, where the adjacency matrix is given by a stochastic block model with two blocks, and the node feature is generated with latent vectors corresponding to the two blocks (Appendix C.1). Both student and teacher network has $L = 2$ branches. We calculate the order parameters $u_l$'s for the student network with the saddle point equations, as well as perform Hamiltonian Monte Carlo (HMC) sampling (Appendix C.5) from the posterior distribution of the student's network weights to generate a distribution of student readout norms squared $a_l^2$. We compare the values of the student branch readout norms to their corresponding teacher branches for the student network with either narrow or wide width. As shown in Figure 1(b), an extremely narrow student network ($N = 4$) learns the teacher's branch readout norms very robustly, while a much wider network ($N = 1024$) fails to learn the teachers' norms and approaches the GP ($N \to \infty$) limit where the two branches are indistinguishable.

**Symmetry breaking and convergence of branches:** We perform theoretical calculations and HMC sampling for the student branch squared readout norms as we vary the student network width $N$ and the prior regularization strength $\sigma_w$. Figure 2(a) shows the statistical average of the student branch squared readout norms, ie. $\langle \|a_l\|^2 \rangle \sigma_w^2 / N$ as a function of the network width $N$, where the branch norms split as the network width gets smaller, which we call symmetry breaking. The symmetry breaking of branch norms from the GP limit to the narrow width limit accompanies the convergence to learning teacher's norms at narrow width for different $\sigma_w$'s as shown in Figure 2(b)(c), supporting Eq. 62.

**Narrow-to-wide width transition:** We can also determine the generalization properties of the student readout lables at both the branch level $f_l$ (Appendix A.3) and overall level $f$ using Eq. 1415. We perform theoretical calculations and HMC sampling of generalization errors as a function of the network width $N$ and regularization strength $\sigma_w$, with results shown in Figure 3 and 4. At narrow width, we expect individual branch $f_l$ to learn the corresponding teacher's branch output $f_l^*$ independently, causing the bias to increase with network width. This is observed for both branches, with a transition from the narrow-width regime to the GP regime. The regularization strength $\sigma_w$ controls the transition window, with larger $\sigma_w$'s leading to sharper transitions. This aligns with our analysis

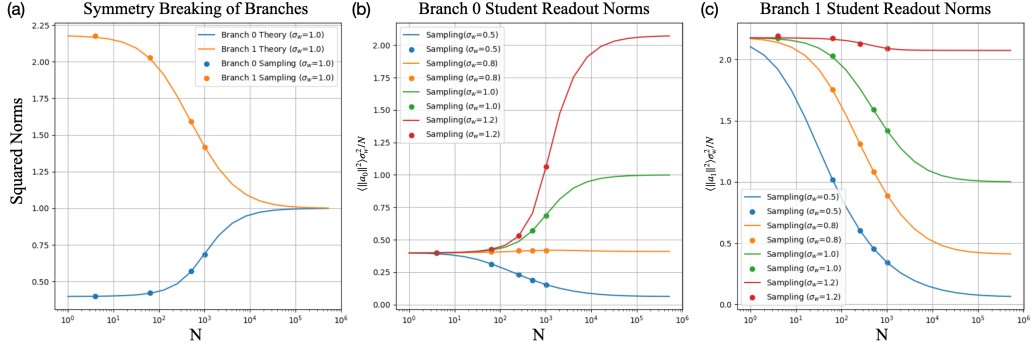

Figure 2: Statistical average of student readout norms squared as a function of network width from theory and HMC sampling, for student-teacher tasks described in Section 4.2. (a): $\langle \|a_l\|^2 \rangle \sigma_w^2 / N$ as a function of network width $N$ for a fixed $\sigma_w$. The branch norms break the GP symmetry as it goes to the narrow width limit. (b)(c): Branch 0 and branch 1 readout norm squared respectively for a range of $\sigma_w$ regularization values. The student branch norms with different regularization strengths all converge to the same teacher readout norm values at narrow width.

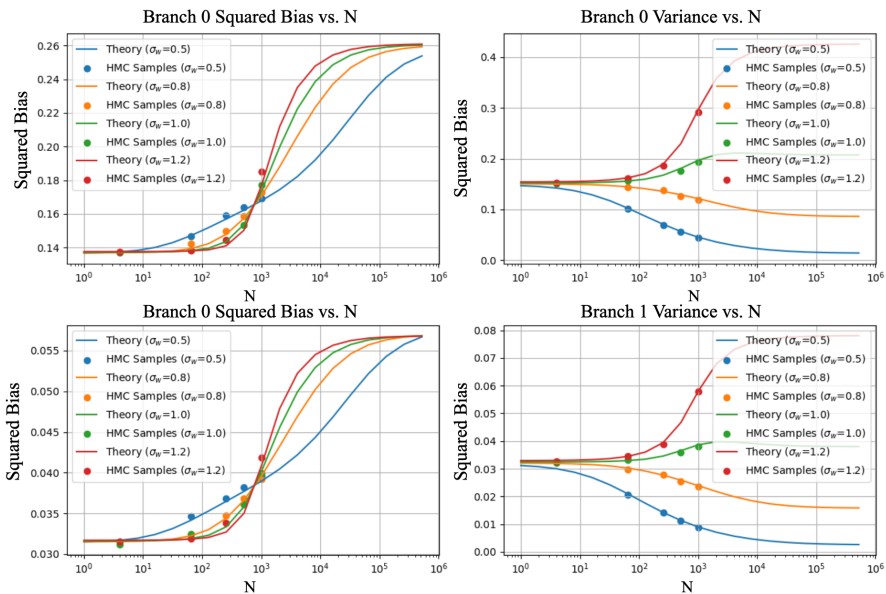

Figure 3: Student network squared bias and variance for individual branches as a function of network width $N$ and regularization strength $\sigma_w$. The mean and variance of branch $l$ readout $f_l^\mu$ for node $\mu$ is calculated in A.3 and the bias and variance for branch $l$ can be infered similarly as Eq. 15. Generalization values are normalized over the average true readout labels.

of the entropic and energetic contributions, where the larger $\sigma_w$ amplifies the distinction between the two terms. In contrast, the variance decreases with network width for small $\sigma_w$'s, resulting in a trade-off between the contributions of bias and variance to overall generalization performance, as shown clearly with the graph of network generalization vs. $N$ with $\sigma_w = 0.5$ in Figure 4.

## 5 BPB-GCN on Cora

We also perform experiments on the Cora benchmark dataset (McCallum et al. (2000)) by training the BPB-GCN with binary node regression, for a range of $L, N, \sigma_w$ values (Appendix C.2 for de-

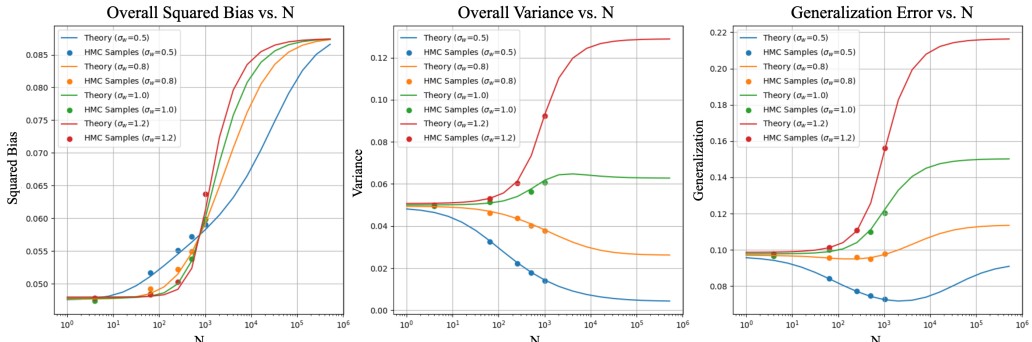

Figure 4: Student network generalization performance as a function of network width $N$ and regularization strength $\sigma_w$. Generalization is normalized over the average true readout labels.

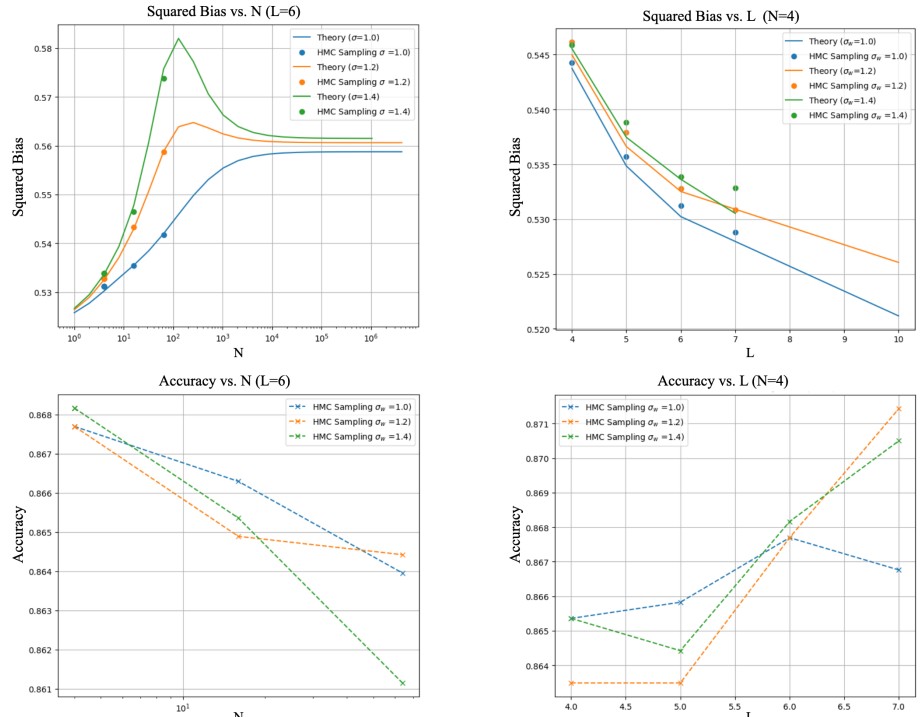

Figure 5: Cora generalization performance vs. network width $N$ and branch number $L$, for various regularization strength $\sigma_w$'s. The accuracy is computed by turning the mean predictor from HMC samples into a class label using its sign.

tails). We observe a similar narrow-to-wide width transition for the bias term. As shown in Figure 5, the bias increases with network width, transitioning to the GP regime, and we observe the trend extending to a potential narrow width limit.[2] Additionally, it is demonstrated that using more branches that involve higher-order convolutions improves performance.

**Convergence of branch readout norms at narrow width** An interesting aspect of the BPB-GCN network is that the branch readout norms converge at the narrow width for different hyperparameters $\sigma_w$ and $L$, reflecting the natural branch importance for the task.

---

[2]In this case, the narrow width limit is hard to demonstrate as the transition window is below realistic minimum of network width.

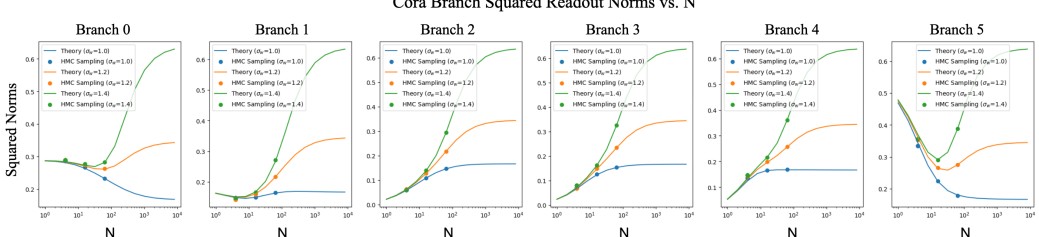

Figure 6: Cora experiment: statistical average of squared readout norms $\langle \|a_l\|^2 \rangle \sigma_w^2/N$ for each branch $l$ as a function of the network width $N$, and regularization strength $\sigma_w$. The BPB-GCN has $L = 6$ branches.

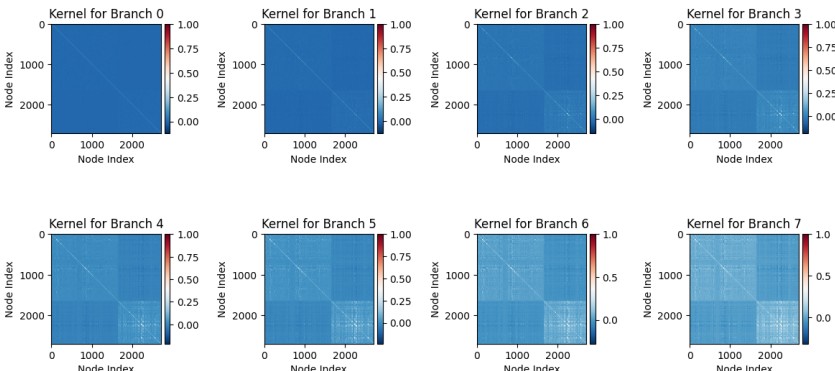

Figure 7: Kernel $K_l = \frac{\sigma_w^2}{N_0} A^l X_0 X_0^T A^l$ for each branch $l$ shown for the first 8 branches on Cora dataset, sorted by node labels. $A$ is the normalized adjacency matrix of the Cora graph, $X_0$ the node feature matrix and $N_0$ the node feature dimension. Initialization variance $\sigma_w^2 = 1$ and total node number $n = 2708$.

As shown in Figure 6, the BPB-GCN with branches $L = 6$ robustly learns the readout norms at narrow width independently of $\sigma_w$'s, consistent with the student-teacher results. This suggests that we can recast the data as generated from a ground-truth teacher network even for real-world datasets. The last branch of the BPB-GCN network has a larger contribution, reflecting the presence of higher-order convolutions in the Cora dataset. From a kernel perspective, increasing branches better distinguish the nodes, as shown in Figure 7. This could explain the selective turn-off of intermediate branches and the increased contribution of the last branch. Our results also indicate that there is no oversmoothing problem at narrow width: as $L$ increases, individual branches can still learn robustly.

Furthermore, the first two branches are learned most robustly at narrow width, as shown in Figure 8, where the branch norms converge for the first two branches even for BPB-GCNs with different $L$. This suggests that the branch importance, as reflected by the norms learned at narrow width, indicates the contribution of the bare data and the first convolution layer.

## 6 DISCUSSION

The findings presented in this paper reveal that BPB-GCNs exhibit unique characteristics in the narrow width limit. Unlike the infinite-width limit, where neural networks behave as Gaussian Process (GP) with task-independent kernels, narrow-width BPB-GCNs undergo significant kernel renormalization. This renormalization leads to breaking of the symmetry between the branches, resulting in more robust and differentiated learning. Our experiments demonstrate that narrow-width BPB-GCNs can retain and, in some cases, improve generalization performance compared to their wider counterparts, particularly in bias-limited scenarios where the regularization effects dominate. There-

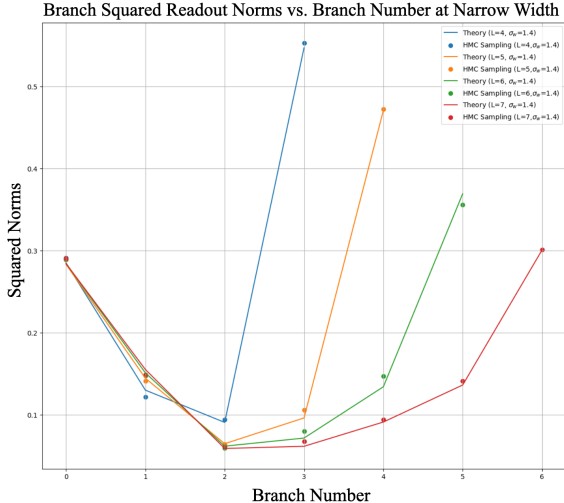

Figure 8: Branch Importance vs. branch number $l$ on Cora. Legends represent different total number of branches $L$. The branch importance is defined as the statistical average of branch squared readout norms $\langle \|a_l\|^2 \rangle \sigma_w^2 / N$ at the narrow width limit; here we take the empirical branch norm values at $N = 4$ and fixed $\sigma_w = 1.4$.

fore, there is a trade-off between the expressive power of GCN Xu et al. (2018); Loukas (2019) and stability Stogin et al. (2020). Additionally, the observed independence of readout norms from architectural hyperparameters suggests that narrow-width BPB-GCNs can capture the intrinsic properties of the data more effectively. These insights suggest potential new strategies for optimizing neural network architectures in practical applications, challenging the traditional emphasis on increasing network width for better performance. Although our work is focused on GCN architectures, our findings are more general for a family of branching networks as shown in Appendix B, and in particular the Bayesian theory with kernel renormalization is shown to be successful for CNN Aiudi et al. (2023) as well as transformers Tiberi et al. (2024). Intriguingly, Vyas et al. (2024) finds that an ensemble of ResNets averaged over random initializations accidentally shows this narrow width effect, for which our findings provide an explanation.

**Limitations:** We demonstrate that the bias robustly decreases at narrow width for the branching networks due to kernel renormalization, which defies the common belief that wider networks are better (Allen-Zhu et al. (2019); Lee et al. (2018)). However, we do find that for real-world scenarios such as BPB-GCN on Cora, and residual-MLP on Cifar10, the generalization error is usually variance-limited. In principle, training an ensemble of branching networks averaged over random initialization helps to reduce the variance significantly such that the narrow-width effect is more pronounced as in Vyas et al. (2024). Furthermore, since our Bayesian setup corresponds to an ensemble of fully trained over-parametrized networks in an offline fashion, the learning is still in the so-called lazy regime with minimal feature learning (Atanasov et al. (2022); Karkada (2024)). We hypothesize that narrower width serves as a regularizer in this regime which might break down as the network is trained with online learning and approaches the rich regime (Yang et al. (2022)) with more feature learning. In that case, it might be possible that even the bias does not decrease at the narrow width.

## 7 CONCLUSION

In conclusion, this paper introduces and investigates the concept of narrow width limits in Bayesian branching networks. Our results indicate that branching networks with significantly narrower widths can achieve better or competitive performance, contrary to common beliefs. This is attributed to effective symmetry breaking and kernel renormalization in the narrow-width limit, which leads to

robust learning. Our theoretical analysis, supported by empirical evidence, establishes a new understanding of how network architecture influences learning outcomes. This work provides a novel perspective on the infinite width limit of neural networks and suggests further research directions in understanding narrow width as a regularization effect.

## ACKNOWLEDGEMENT

We acknowledge support of the Swartz Foundation, the Kempner Institute for the Study of Natural and Artificial Intelligence at Harvard University and the Gatsby Charitable Foundation. This material is partially based upon work supported by the Center for Brains, Minds and Machines (CBMM), funded by NSF STC award CCF-1231216. We have benefitted from helpful discussions with Alexander van Meegen, Lorenzo Tiberi and Qianyi Li.

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

# A  DETAILS ON THEORY OF BPB-GCN

## A.1  SUMMARY OF NOTATIONS

### Hyperparameters and Dimensions

| | |
|---|---|
| $P$ | Number of training nodes |
| $n$ | Total number of nodes for a graph |
| $N_0$ | Input node feature dimension |
| $N$ | BPB-GCN hidden layer width |
| $L$ | Total number of branches |
| $\sigma_w$ | $L_2$ prior regularization strength |
| $T$ | Temperature |
| $\alpha_0 = \dfrac{P}{LN_0}$ | Network capacity |
| $\alpha = \dfrac{P}{N}$ | Width ratio |

### Network Architecture and Input/Output

| | |
|---|---|
| $\hat{A} \in \mathbb{R}^{n \times n}$ | Adjacency matrix |
| $A \in \mathbb{R}^{n \times n}$ | Normalized adjacency matrix by its degree matrix $D$ |
| $X \in \mathbb{R}^{n \times N_0}$ | Input node feature matrix |
| $G = (X, A)$ | Graph |
| $W^{(l)} \in \mathbb{R}^{N_0 \times N}$ | Hidden layer weight for branch $l$ |
| $a^{(l)}, a_l \in \mathbb{R}^N$ | Readout vector for branch $l$ |
| $\Theta_l = (W_l, a_l)$ | Collection of parameters for branch $l$ |
| $h_l^\mu \in \mathbb{R}^N$ | Activation vector for branch $l$ and node $\mu$ |
| $f_l^\mu \in \mathbb{R}$ | Readout prediction for branch $l$ and node $\mu$ |
| $f^\mu \in \mathbb{R}$ | Overall readout prediction for node $\mu$ |
| $y^\mu \in \mathbb{R}$ | Overall node label for node $\mu$ |
| $H_l(W_l) \in \mathbb{R}^{P \times N}$ | Activation feature matrix for branch $l$ |
| $Y \in \mathbb{R}^P$ | Training node labels |
| $F \in \mathbb{R}^P$ | Readout predictions |

### Statistical Theory

| | |
|---|---|
| $E(\Theta; G, Y)$ | Energy loss function |
| $Z$ | Partition function |
| $I \in \mathbb{R}^{P \times P}$ | Identity matrix |
| $H(W)$ | Hamiltonian after integrating out readout $a_l$'s |
| $u \in \mathbb{R}^L$ | Order parameters as saddle point solution |
| $u_l \in \mathbb{R}$ | Order parameter for branch $l$ |
| $H(u)$ | Hamiltonian as a function of order parameters |
| $S(u)$ | Entropy as a function of order parameters |
| $E(u)$ | Energy as a function of order parameters |
| $r_l$ | Mean squared readout |
| $\mathrm{Tr}_l$ | Variance-related of readout |
| $\langle x \rangle$ | Statistical average of any quantity $x$ over the posterior distribution |

### Kernels

| | |
|---|---|
| $K(W) \in \mathbb{R}^{P \times P}$ | Hidden layer weight dependent overall kernel |
| $K_l$ | Branch $l$ kernel |
| $K \in \mathbb{R}^{P \times P}$ | Overall kernel averaged over $W$'s, $K = \sum_l \frac{u_l K_l}{L}$ |
| $k_l^\nu \in \mathbb{R}^{P \times 1}$ | Branch $l$ kernel column for the $P$ training nodes against test node $\nu$ |
| $k^\nu \in \mathbb{R}^{P \times 1}$ | Overall kernel column for the $P$ training nodes against test node $\nu$ |
| $\mid_P$ | Kernel restricted to the $P$ training nodes against $P$ training nodes |
| $\mid_{(P,\nu)}$ | Kernel restricted to $P$ training nodes against the test node $\nu$ |
| $\mid_{(\nu,\nu)}$ | Kernel restricted to the test node $\nu$ against test node $\nu$ |

### Student-teacher Setup

| | |
|---|---|
| $Y_l^* \in \mathbb{R}^P$ | Teacher network readout prediction for branch $l$ for $P$ nodes |
| $Y^* \in \mathbb{R}^P$ | Teacher network overall readout for $P$ nodes |
| $W_l^* \in \mathbb{R}^{N_0 \times N}$ | Teacher hidden layer weight for branch $l$ |
| $a_l^* \in \mathbb{R}^N$ | Teacher readout vector for branch $l$ |
| $\beta_l^2$ | Teacher readout variance |
| $\sigma_t^2$ | Teacher hidden layer weight variance |

### A.2   KERNEL RENORMALIZATION

Following similar derivations as the first kernel renormalization work Li & Sompolinsky (2021), we will integrate out the weights in the partition function $Z = \int d\theta \exp(-E(\Theta)/T)$, from the readout layer weights $a_l$'s to the hidden layer weights $W_l$'s and arrive at an effective Hamiltonian shown in the main text.

First, we linearize the energy in terms of $a_l$'s and transform the integral by introducing the auxiliary variables $t^\mu, \mu = 1, \ldots, P$.

$$Z = \int d\Theta \int \prod_{\mu=1}^{P} dt_\mu \exp\left[ -\frac{1}{2\sigma_w^2}\Theta^\top\Theta - \sum_{\mu=1}^{P} it_\mu \left( \frac{1}{\sqrt{LN}} \sum_{i=1}^{N} \sum_{l=0}^{L-1} a_i^{(l)} h_i^\mu(G) - Y^\mu \right) - \frac{T}{2}t^\top t \right]$$

(22)

Now we can integrate out $a_l$'s as they are linearized and the partition function becomes

$$Z = \int DW e^{-H(W)},$$

(23)

with effective Hamiltonian

$$H(W) = \frac{1}{2\sigma_w^2} \sum_{l=0}^{L-1} \text{Tr} W_l^T W_l + \frac{1}{2} Y^T (K(W) + TI)^{-1} Y + \frac{1}{2} \log \det(K(W) + TI),$$

(24)

where

$$K(W) = \frac{1}{L} \sum_l \frac{\sigma_w^2}{N_0} (H_l(W_l) H_l(W_l)^T)|_P$$

(25)

is the $P \times P$ kernel matrix dependent on the observed $P$ nodes with node features $H_l = A^l X W_l$ and denote $|_P$ as the matrix restricting to the elements generated by the training nodes.

Now we perform the integration on $W_l$'s and introduce the auxiliary variables $t^\mu$'s again using the Gaussian trick and get

$$Z = \int \prod_{l=0}^{L-1} dW_l \int dt \exp\left[ -\frac{1}{2}t^T(K(W) + TI)t - \frac{1}{2\sigma_w^2} \sum_l \text{Tr}(W_l^T W_l) + it^T Y \right]$$

(26)

$$= \int dt \exp\left[ it^T Y + G(t) - \frac{T}{2}t^T t \right],$$

(27)

where $G(t)$ is in terms of the kernel averaged over the Gaussian measure

$$G(t) = \log \left\langle \exp\left( -\frac{1}{2N}t^T K(W)t \right) \right\rangle_W$$

(28)

Writing out the integral explicitly, we have

$$G(t) = \log \int \prod_{l=0}^{L-1} DW_l \exp\left( -\frac{1}{2N} \sum_{\mu,\nu} t^\mu t^\nu \sum_{j,l} \frac{\sigma_w^2}{N_0 L} (\sum_{\mu',i} A_{\mu,\mu'}^l W_{ij}^l x_i^{\mu'})(\sum_{\nu',i'} A_{\nu,\nu'}^l W_{i'j}^l x_{i'}^{\nu'}) \right)$$

(29)

where $DW_l$ is the Gaussian measure

$$DW_l = \prod_{i,j} e^{-W_{l,i,j}^2/2\sigma_w^2} dW_{l,i,j}$$

(30)

By recognizing that this is just a product of the same $N$ integrals on $N_0$ dimensional weight vector $\vec{W}_l^j$ for a fixed $j$ neuron, we get

$$G(t) = N \log \int \prod_{l=0}^{L-1} d\vec{W}_l^j \exp\left( \sum_{l,i,i'} (-\frac{1}{2N} \sum_{\mu,\nu} t^\mu t^\nu \frac{\sigma_w^2}{N_0 L} \xi_{l,i}^\mu \xi_{l,i'}^\nu + \frac{1}{2\sigma_w^2} \delta_{i,i'}) W_{i,l}^j W_{i',l}^j \right)$$

(31)

where $\xi_{l,i}^\mu = \sum_{\mu'} A_{\mu,\mu'}^l x_i^{\mu'}$ is the $l$th branch convolution on node $\mu$. Since the network is linear, we are able to perform the Gaussian integration exactly and get

$$G(t) = -\frac{N}{2} \sum_l \log(\det(I_{i,i'} + \frac{\sigma_w^2}{N_0 L} t^\mu \xi_{i,l}^\mu \xi_{i',l}^\nu t^\nu))$$

(32)

where the determinant is over the $i, i'$ matrix, and thus performing the determinant we arrive at

$$G(t) = -\frac{N}{2} \sum_l \log(1 + t^T \frac{K_l}{L} t)), \tag{33}$$

where $K^l_{\mu,\nu} = \frac{\sigma^2_w}{N} \sum_i t^\mu \xi^\mu_{i,l} \xi^\nu_{i,l}$ is the Gaussian process kernel for branch $l$.

Next, we insert $L$ delta functions $\int \prod_l dh_l \delta(h_l - t^T K_l t)$ to enforce the identity $G(t) = \sum_l -\frac{N}{2} \log(1 + h_l)$ and use the Fourier representation of it with auxiliary variables $u_l$ to get:

$$Z = \int \prod_{l=0}^{L-1} dh_l du_l dt \exp\left(it^T Y - \sum_l \frac{N}{2} \log(1 + h_l) + \sum_l \frac{N}{2\sigma^2_w} u_l h_l - \frac{1}{2} t^T \left(\sum_l \frac{1}{L} u_l K_l + TI\right) t\right)$$

$$= \int \prod_{l=0}^{L-1} dh_l du_l \exp\left(-\sum_l \frac{N}{2} \log(1 + h_l) + \sum_l \frac{N}{2\sigma^2_w} u_l h_l \frac{1}{2} Y^T \left(\sum_l \frac{1}{L} u_l K_l + TI\right)^{-1} Y\right) \tag{34}$$

where

$$K_l = \frac{\sigma^2_w}{N_0} [A^l X X^T A^{lT}]|_P \tag{35}$$

is the input kernel for branch $l$. Now as $N \to \infty$ and $\alpha = \frac{P}{N}$ fixed, we can perform the saddle point approximation and get the saddle points for $h_l$ as

$$1 + h_l = \frac{\sigma^2_w}{u_l} \tag{36}$$

Plugging this back to the equation, we get

$$Z = \int \Pi_l du_l e^{-H_{eff}(u)}, \tag{37}$$

with the effective Hamiltonian

$$H_{eff}(u) = S(u) + E(u), \tag{38}$$

where we call $S(u)$ the entropic term

$$S(u) = -\sum_l \frac{N}{2} \log u_l + \sum_l \frac{N}{2\sigma^2_w} u_l \tag{39}$$

and $E(u)$ the energetic term

$$E(u) = \frac{1}{2} Y^T \left(\sum_l \frac{1}{L} u_l K_l + TI\right)^{-1} Y + \frac{1}{2} \log \det\left(\sum_l \frac{1}{L} u_l K_l + TI\right) \tag{40}$$

Therefore, after integrating out $W_l$, the effective kernel is given by

$$K = \sum_l \frac{1}{L} u_l K_l, \tag{41}$$

and the saddle point equations for $u_l$'s are determined by

$$N(1 - \frac{u_l}{\sigma^2_w}) = -Y^T (K + TI)^{-1} \frac{u_l K_l}{L} (K + TI)^{-1} Y + \text{Tr}[K^{-1} \frac{u_l K_l}{L}], \tag{42}$$

where we call

$$r_l = Y^T (K + TI)^{-1} \frac{u_l K_l}{L} (K + TI)^{-1} Y \tag{43}$$

and

$$\text{Tr}_l = \text{Tr}[K^{-1} \frac{u_l K_l}{L}] \tag{44}$$

As we will show later, these represent the mean and variance of the readout norm squared respectively. In the $T = 0$ case, the saddle point equation becomes

$$N(1 - \frac{u_l}{\sigma^2_w}) = -Y^T K^{-1} \frac{u_l K_l}{L} K^{-1} Y + \text{Tr}[K^{-1} \frac{u_l K_l}{L}] \tag{45}$$

### A.3 PREDICTOR STATISTICS AND GENERALIZATION

We can get the predictor statistics of each branch readout $y_l^\nu(G)$ on a new test node $\nu$ by considering the generating function:

$$
\begin{aligned}
Z(\eta_1, \ldots, \eta_L) = \int D\Theta \exp \Bigg\{ &-\frac{\beta}{2} \sum_\mu (f^\mu(G; \Theta) - y^\mu)^2 \\
&+ \sum_l i\eta_l \frac{1}{\sqrt{NL}} \sum_i a_i^{(l)} h_i^{(l),\nu}(G, W_l) - \frac{T}{2\sigma_w^2} \Theta^T \Theta \Bigg\}
\end{aligned}
\tag{46}
$$

Therefore, by taking the derivative with respect to each $\eta_l$, we arrive at the statistics for $y_l(x)$ as:

$$
\langle f_l^\nu(G) \rangle = \partial_{i\eta_l} \log Z \big|_{\vec{\eta}=0}
\tag{47}
$$

$$
\langle \delta f_{l,\nu}^2(G) \rangle = \partial_{i\eta_l}^2 \log Z \big|_{\vec{\eta}=0}
\tag{48}
$$

After integrating out the weights $\Theta$ layer by layer, we have:

$$
\begin{aligned}
Z(\eta_1, \ldots, \eta_L) = \int \Pi_l du_l \ \exp \Bigg\{ &\sum_l \left( \frac{N}{2} \log u_l - \frac{N}{2\sigma_w^2} u_l \right) \\
&+ \frac{1}{2} \left( iY + \sum_l \frac{1}{L} \eta_l u_l k_l^\nu \right)^T \left( \sum_l \frac{1}{L} u_l K_l + TI \right)^{-1} \left( iY + \sum_l \eta_l \frac{1}{L} u_l k_l^\nu \right) \\
&- \frac{1}{2} \log \det \left( \sum_l \frac{1}{L} u_l K_l + TI \right) - \frac{1}{2} \sum_l \eta_l^2 \frac{1}{L} K_l^{\nu,\nu} \Bigg\}.
\end{aligned}
\tag{49}
$$

Here

$$
k_l^\nu = \frac{\sigma_w^2}{N_0} [A^l X X^T A^l] \big|_{(P,\nu)}
\tag{50}
$$

is the $P \times 1$ column kernel matrix for test node $\nu$ and all training nodes, and

$$
K_l^{\nu,\nu} = \frac{\sigma_w^2}{N_0} [A^l X X^T A^l] \big|_{(\nu,\nu)}
\tag{51}
$$

is the single matrix element for the test node. Therefore, eventually, we have:

$$
\langle f_l^\nu \rangle = \frac{u_l k_{l,\nu}^T}{L} (K + TI)^{-1} Y
\tag{52}
$$

and

$$
\langle \delta f_{l,\nu}^2 \rangle = \frac{u_l K_l^{\nu,\nu}}{L} - \frac{u_l k_{l,\nu}^T}{L} (K + TI)^{-1} \frac{u_l k_{l,\nu}}{L}
\tag{53}
$$

The predictor statistics of the overall readout $f = \sum_l f_l$ is given by:

$$
\langle f^\nu(G) \rangle = \sum_l \frac{u_l k_{l,\nu}^T}{L} (K + TI)^{-1} Y = k_\nu^T (K + TI)^{-1} Y
\tag{54}
$$

$$
\langle \delta f(G)_\nu^2 \rangle = \sum_l u_l K_l^{\nu,\nu} - \sum_{l,l'} u_l k_{l,\nu}^T (K + TI)^{-1} u_{l'} k_{l',\nu} = K_{\nu,\nu} - k_\nu^T (K + TI)^{-1} k_\nu
\tag{55}
$$

### A.4 STATISTICS OF BRANCH READOUT NORMS

From the partition function Eq.22, we can relate the mean of readout weights $a_l$ to the auxiliary variable $t$ by

$$\langle a_l \rangle_W = -i \frac{\sigma_w^2}{\sqrt{N}} \Phi_l^T \langle t \rangle = -\frac{\sigma_w^2}{\sqrt{NL}} \Phi_l^T (K + TI)^{-1} Y, \tag{56}$$

where $\Phi_l$ is the node feature matrix for the hidden layer nodes. We have

$$\langle a_l^T \rangle \langle a_l \rangle = \sigma_w^2 Y^T (K + TI)^{-1} \frac{u_l K_l}{L} (K + TI)^{-1} Y = r_l \sigma_w^2 \tag{57}$$

We can calculate the second-order statistics of $a_l$: the variance is

$$\langle \delta a_l^T \delta a_l \rangle = \sigma_w^2 \mathrm{Tr}(I + \frac{\sigma_w^2 \beta}{NL} \Phi_l \Phi_l^T)^{-1} = \sigma_w^2 (N - \mathrm{Tr}(K + TI)^{-1} \frac{u_l K_l}{L}) = \sigma_w^2 (N - \mathrm{Tr}_l) \tag{58}$$

Therefore,

$$\langle a_l^2 \rangle = \langle \delta a_l^T \delta a_l \rangle + \langle \delta a_l^T \delta a_l \rangle = N\sigma_w^2 + \sigma_w^2 r_l + 1 - \sigma_w^2 \mathrm{Tr}_l = Nu_l \tag{59}$$

Therefore, we have proved the main text claim that the order parameter $u_l$'s are really the mean squared readout norms of the branches.

## B THE NARROW WIDTH LIMIT FOR GENERAL ARCHITECTURES

In this section, we demonstrate that our results on the narrow-width limit can be generalized to any network with branching structures, supported by both theoretical and empirical evidence.

Consider a general branching network, with $L$ independent branches each with hidden layer weight $W_l$ and readout weight $a_l$, ie.

$$f(x; \Theta) = \sum_{l=0}^{L-1} \frac{1}{\sqrt{L}} f_l(x; \Theta_l) = \sum_l \frac{1}{\sqrt{NL}} a_l \phi_l(x; W_l), \tag{60}$$

where $\phi_l(x; W_l)$ represents any feed-forward function on the dataset $X$, ie. in matrix form the branch $l$ readout is $F_l = \phi_l(\frac{1}{\sqrt{N_0}} X W_l)$. As an example, $\phi_l(X W_l) = \frac{1}{\sqrt{N_0}} A^l X W_l$ represents $l$ convolutions on the node features in the BPB-GCN architecture. $\phi_l(X W_l) = \frac{1}{\sqrt{N_0}} A_l X W_l$ can also represent an attention head $l$ for attention-networks with frozen attention $A_l$, as well as a kernel convolution for patch $l$ in the CNN architecture. In the residual-MLP example below, $\phi_l$ represents either the linear or the ReLu branch.

With the general branching network, we can similarly consider the posterior distribution in the Bayesian setup as Eq. 4 with squared error as the likelihood term and $L_2$ regularization as the prior term. Provided the network is *linear*, the partition function Eq.5 can be integrated in a similar fashion and arrive at the saddle point equation

$$N(1 - \frac{u_l}{\sigma_w^2}) = -Y^T (K + TI)^{-1} \frac{u_l K_l}{L} (K + TI)^{-1} Y + \mathrm{Tr}[K^{-1} \frac{u_l K_l}{L}], \tag{61}$$

with $K = \sum_l u_l K_l / L$ and $K_l$ corresponding to the NNGP kernel (Lee et al. (2018)). In fact, even with non-linear activations, we conjecture that the kernel renormalization still holds with the order parameters $u_l$ that satisfies the saddle point equation, and the posterior weight statistics calculated with the renormalized kernel agrees with the experiments as empirically shown in the residual-MLP section.

### B.1 THE GENERALIZED EQUIPARTITION THEOREM

With the general branching network, we show the narrow width limit result for the student-teacher setup. Specifically we prove the following

**The generalized equipartition theorem** : For the general branching network Eq. 60, where the teacher network width is $N_t$ with $W^*_{l,ij} \sim \mathcal{N}(0, \sigma_t^2)$ and $a^*_{i,l} \sim \mathcal{N}(0, \beta_l^2)$ and the student network with layer width $N$ in the Bayesian regression setup of Eq. 4, at $T = 0$ temperature, in the limit $N_t \to \infty$ and $\alpha = P/N \to \infty$, there exists a solution to the saddle point equations 12

$$u_l = \sigma_t^2 \beta_l^2 / \sigma_w^2. \tag{62}$$

*Proof*: The proof is exactly the same as before, except for changing $K_l$ as a general NNGP kernel that corresponds to the feature map $\phi_l$ for branch $l$.

## B.2 RESIDUAL-MLP

In the following, we show a simple 2-layer residual network as an example for the general branching networks, with a linear branch $\phi_0(x) = \frac{1}{\sqrt{N_0}} W_0 x$ and a ReLU branch $\phi_1(x) = \frac{1}{\sqrt{N_0}} \mathrm{ReLu}(W_1 x)$. Therefore, the overall kernel is $K = \frac{1}{2}(u_0 K_0 + u_1 K_1)$, with the NNGP kernels $K_l$ given by

$$K_0 = K_{linear} = \frac{\sigma^2}{N_0} X_0 X_0^T \tag{63}$$

and the relu kernel (Cho & Saul (2009))

$$K_1(x, x') = K_{relu}(x, x') = \frac{1}{2\pi} \sqrt{K_0(x, x) K_0(x', x')} J(\theta), \tag{64}$$

where

$$J(\theta) = \sin(\theta) + (\pi - \theta) \cos(\theta) \tag{65}$$

and

$$\theta = \cos^{-1}\left(\frac{K_0(x, x')}{\sqrt{K_0(x, x) K_0(x', x')}}\right) \tag{66}$$

For the student-teacher task (details in Appendix C.3 ), we observe the same narrow-width limit phenomenon as in the BPB-GCN case, where each student branch robustly learns the teacher's corresponding branch but approaches the same GP limit as the network width gets larger.

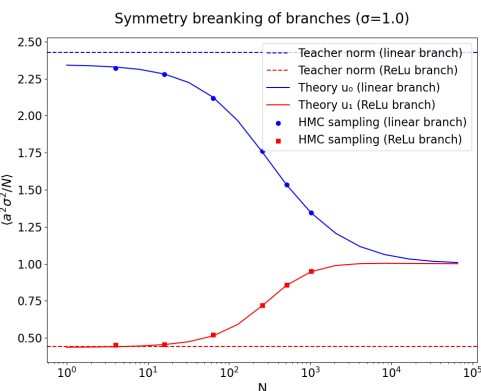

Figure 9: Statistical average of student readout norms squared as a function of network width from theory and HMC sampling, for the residual-MLP student-teacher task, with fixed $\sigma_w = 1$.

As shown in Figure 9 and 10, there is a symmetry breaking of branches from the GP-limit to the narrow-width limit, with the ReLu branch and the linear branch learning their corresponding teacher branch norms as $N$ decreases. Furthermore, narrower network performs better in terms of the bias and larger $\sigma_w$ corresponds to a shorter transition window from the GP-limit to the narrow-width limit, similar to the BPB-GCN case. As shown in Figure 11 and 12, the trade-off between the bias and variance contribution is more pronounced for small $\sigma_w$'s, usually with an optimal width for the generalization error. These results support our generalized equipartition theorem and demonstrates that the narrow width limit is a general phenomenon for branching networks.

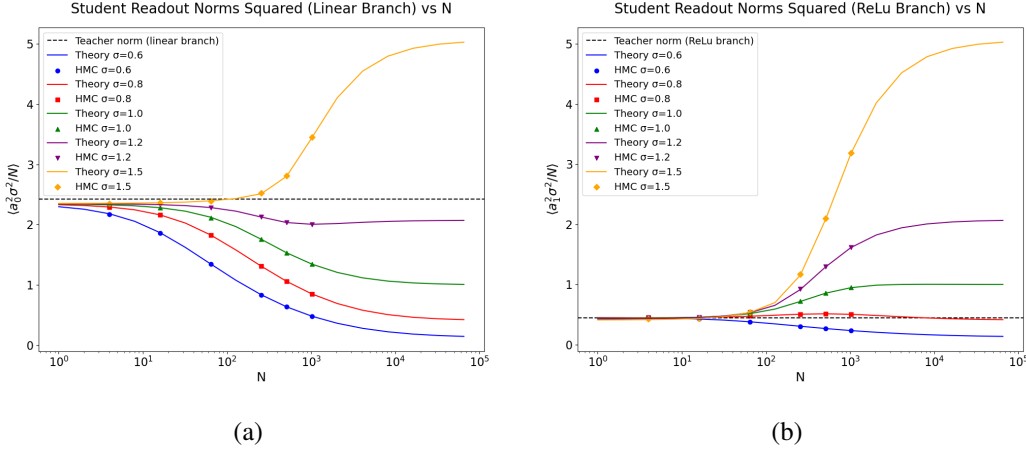

(a)            (b)

Figure 10: $\langle \|a_l\|^2 \rangle \sigma_w^2 / N$ as a function of network width $N$ for a range of $\sigma_w$ values. (a) Linear branch; (b) Relu branch. The student branch norms with different regularization strengths all converge to the same teacher readout norm values at narrow width.

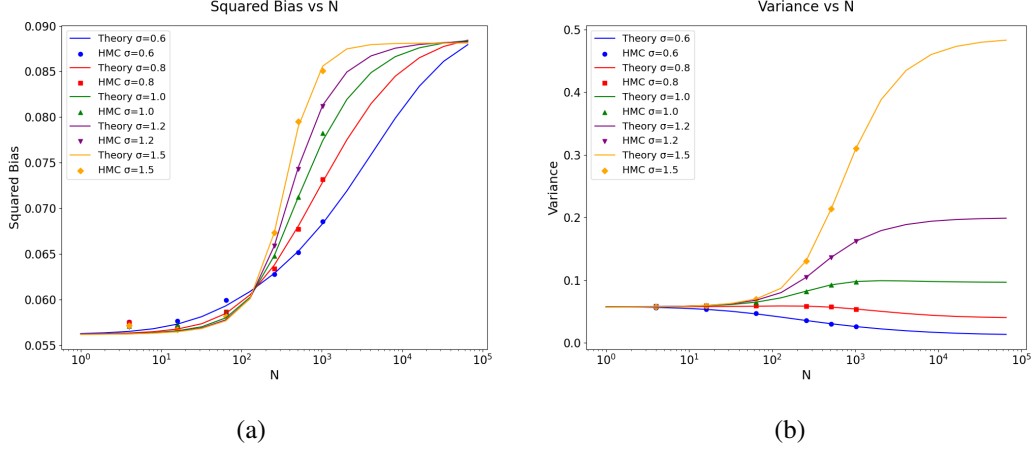

(a)            (b)

Figure 11: Student residual-MLP network generalization performance as a function of network width $N$ and regularization strength $\sigma_w$. (a) Bias; (b) Variance.

### B.3 RESIDUAL-MLP ON CIFAR10

We also perform theory and experiments on the Cifar10 (Krizhevsky et al. (2009)) dataset for a binary classification task (translated to Bayesian regression), with details elaborated in Appendix C.4. Similarly, there exists symmetry breaking of branches and the branch norms converge at the narrow width for different $\sigma_w$'s, as shown in Figure 13,14. Interestingly, the ReLu branch completely dominates at the narrow width, which means the ReLu branch is much more important for the residual-MLP when the training sample size is much bigger than the network width.

Furthermore, we also show in Figure 15,16(a) that the dominance of ReLu branch at narrow width drives the bias to decrease, albeit overall genralization is dominated by the variance. Using the average of the label predictors over different experimental samples, we show that the test accuracy indeed decreases with larger network width $N$ in Figure 16(b).

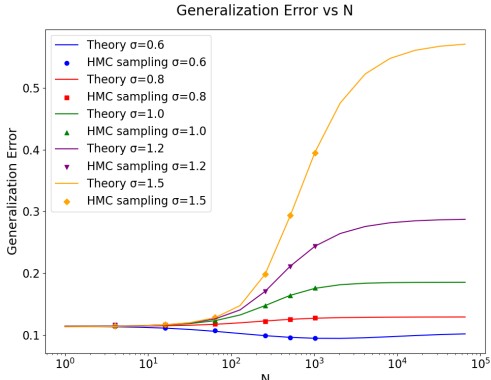

Figure 12: Overall generalization as a sum of bias and variance in Figure 11.

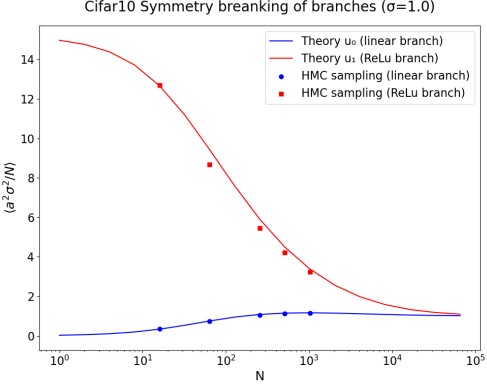

Figure 13: Statistical average of student readout norms squared as a function of network width from theory and HMC sampling, for the Cifar10 dataset with residual-MLP, with fixed $\sigma_w = 1$.

## C    EXPERIMENTAL DETAILS

### C.1    STUDENT-TEACHER CSBM

For the student-teacher task, we use the contextual stochastic block model introduced by Deshpande et al. (2018) to generate the graph $G$. The adjacency matrix is given by

$$A_{ij} = \begin{cases} 1 & \text{with probability } p = c_{in}/n, \text{ if } i, j \leq n/2 \\ 1 & \text{with probability } p = c_{in}/n, \text{ if } i, j \geq n/2 \\ 1 & \text{with probability } q = c_{out}/n, \text{ otherwise} \end{cases} \tag{67}$$

where

$$c_{in,out} = d \pm \sqrt{d}\lambda \tag{68}$$

$d$ is the average degree and $\lambda$ the homophily factor.

The feature vector $\vec{x}^\mu$ for a particular node $\mu$ is given by

$$\vec{x}_\mu = \sqrt{\frac{\mu}{n}} y^\mu \vec{u} + \vec{\xi}_\mu, \tag{69}$$

where

$$\vec{u} \sim \mathcal{N}(0, I_{N_0}), \vec{\xi}_\mu \sim \mathcal{N}(0, I_{N_0}) \tag{70}$$

In the experiment, we use $N_0 = 950$, $d = 20$, $\lambda = 4$ and $\mu = 4$. The teacher network parameters are variance $\sigma_t^2 = 1$, width $N_t = 1024$, branch norms variance $\beta_0^2 = 0.4$, $\beta_1^2 = 2$ for individual

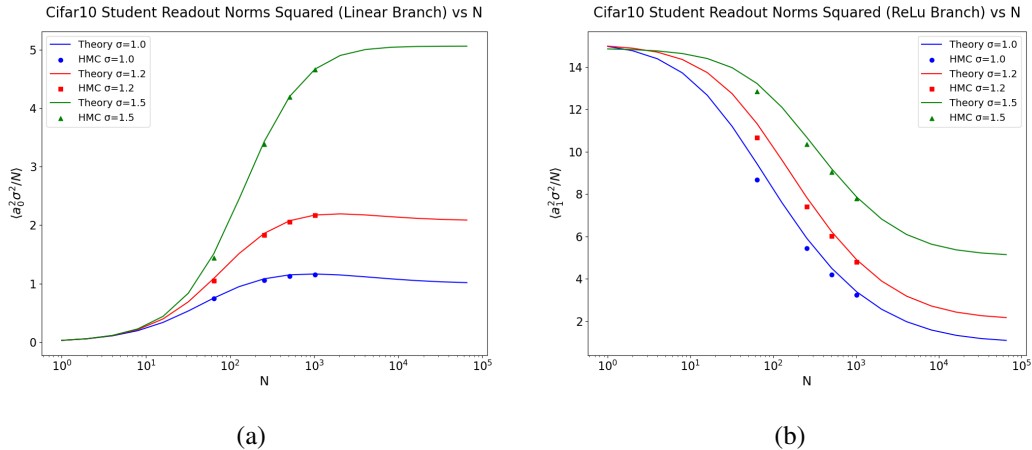

Figure 14: Cifar10 $\langle\|a_l\|^2\rangle\sigma_w^2/N$ as a function of network width $N$ for a range of $\sigma_w$ values. (a) Linear branch; (b) Relu branch. The student branch norms with different regularization strengths all converge to the same teacher readout norm values at narrow width.

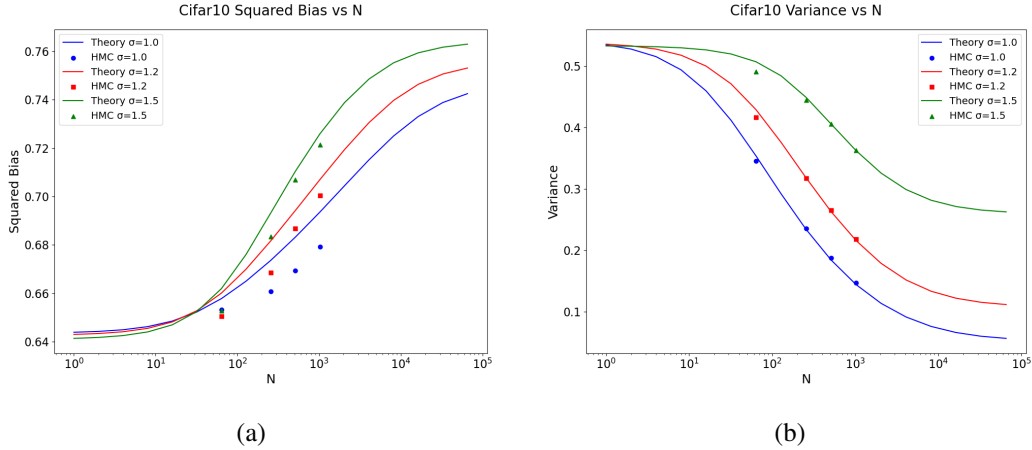

Figure 15: Residual-MLP generalization performance on Cifar10 as a function of network width $N$ and regularization strength $\sigma_w$. (a) Bias; (b) Variance.

element of the readout vector $a_l$. We used total number of nodes $n = 2600$ with train ratio 0.65 for semi-supervised node regression. Temperature $T = 0.0005\sigma_w^2$ for each $\sigma_w$ value in $\sigma_w = \{0.5, 0.8, 1.0, 1.2\}$.

## C.2 CORA

For the Cora dataset, we use a random split of the data (total nodes $n$ =2708) into 21% as training set and 79% as test set. We group the classes $(1, 2, 4)$ into one group and the rest for the other group for binary node regression, with labels as $\pm1$'s. The Bayesian theory and HMC sampling follows the same design as in the student-teacher setup. We use temperature $T = 0.01$ for both theory and sampling as the sampling becomes more difficult for smaller temperature. This explains the discrepancy of the GP limit bias for different $\sigma_w$ values in $\sigma_w = \{1.0, 1.2, 1.4\}$ .

## C.3 STUDENT-TEACHER RESIDUAL-MLP

For the student-teacher task of the residual MLP, we use the data matrix $X$ generated from i.i.d unit Gaussian distribution, with input dimension $N_0 = 1024$, training sample size $P = 1280$ and

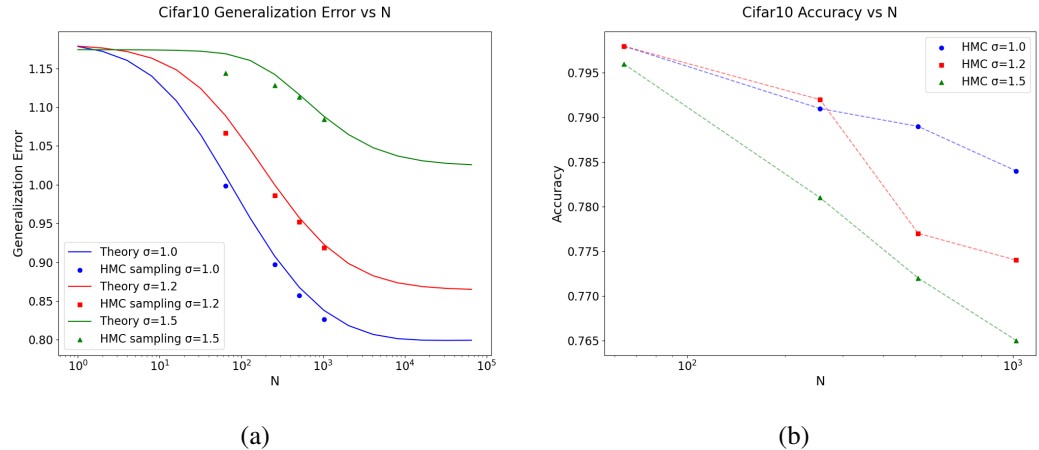

Figure 16: (a) Overall generalization error for residual-MLP on Cifar10. (b) Test accuracy on Cifar10, calculated using the class label by averaging the predictions from HMC samples.

test sample size $P_{test} = 320$. The teacher network weights are also generated with i.i.d Gaussian, with the hidden layer width $N_t = 1024$, hidden layer weight variance $\sigma_w^2 = 1$, and readout branch variance $\beta_l^2 = (2.4, 0.4)$ for the two branches. The temperature is set as $T = 0.001\sigma_w^2$ for a range of $\sigma_w$ values in $\sigma_w = \{0.6, 0.8, 1.0, 1.2, 1.5\}$.

### C.4 CIFAR10

For the Cifar10 dataset, we filter the category of airplane and that of cat for the regression task. We use a random split of training sample size $P = 1000$ and test sample size $P_{test} = 1000$. The input data is only from the red channel with input dimension $N_0 = 1024$. The temperature is set at $T = 0.005\sigma_w^2$ for a range of $\sigma_w$ values in $\sigma_w = \{1.0, 1.2, 1.5\}$. Within the compute budget, we observe significant sample correlations, which has a large impact on measuring the bias term. This is why we observe relatively large deviation of the bias results from theory in Figure 15(a).

### C.5 HAMILTONIAN MONTE CARLO

The sampling experiments in the paper are all done with Hamiltonian Monte Carlo (Betancourt (2017); Neal (2012); Chandra et al. (2021)) simulations, a popular method for sampling a probability distribution. HMC has faster convergence to the posterior distribution compared to Langevin dynamics. We used Numpyrho to set up chains and run the simulations on the GPU cluster. All experiments are within 20 GPU hours budget on Nvidia A100-40G. Due to memory constraint, we only sampled up to $N = 1024$ hidden layer width for the student-teacher CSBM experiment and $N = 64$ for the Cora experiment. Since we mainly aim to demonstrate the narrow width effect in this paper, this suffices the purpose.

