# OpenReview forum: "When narrower is better: the narrow width limit of Bayesian parallel branching neural networks"
_ICLR.cc/2025/Conference — ICLR 2025 Poster_

### Official Review · Reviewer_fRMt · 2024-10-26

**Soundness:** 3
**Presentation:** 3
**Contribution:** 3
**Rating:** 6
**Confidence:** 4

**Summary:**

This paper challenges the common belief that wider networks generalize better. The authors focus on the narrow-width regime, where the network's width is significantly smaller than the number of training samples, and demonstrate that BPB-GNNs can achieve comparable or even superior performance to wide networks in bias-limited scenarios. This improved performance is attributed to symmetry-breaking effects in kernel renormalization, which enhances the robustness of each branch in the network. The paper provides a theoretical framework for understanding the narrow-width limit and validates it with empirical results, including experiments on the Cora dataset. The findings suggest that narrow-width BPB-GNNs can offer efficient alternatives to wide networks, highlighting their potential for optimizing network design in various applications.

**Strengths:**

- It is interesting to see the authors introduce the Bayesian Parallel Branching GNN (BPB-GNN) architecture, which differs from previous neural tangent kernel research.

- The analysis presented in Sections 3 and 4 is comprehensive.

- A statistical physics view is applied to the graph neural network, which is novel and interesing.

**Weaknesses:**

- One main concern is why this work focuses on graph neural networks. It seems that the analysis could be applied to other neural networks, such as MLPs and CNNs. It is hard to see the uniqueness of graph neural networks here.

- The rigor of this work is insufficient. For instance, the definition of $<>$ in Eq. 13 was not introduced.

- To strengthen the experimental section, it would be valuable to include additional real-world datasets (e.g., PubMed, Citeseer) (or tasks beyond node classification).

- To our knowledge, this is the first work to provide a tight bound on the generalization error for GNNs with residual-like structures. However, this may be an overclaim, as other works exist. It would be better to compare with the PAC-Bayes bound for Graph Neural Networks [1].

- A missing reference: [2]

[1] Liao, Renjie, Raquel Urtasun, and Richard Zemel. "A pac-bayesian approach to generalization bounds for graph neural networks." arXiv preprint arXiv:2012.07690 (2020).

[2] Huang, W., Li, Y., Du, W., Yin, J., Da Xu, R.Y., Chen, L. and Zhang, M., 2021. Towards deepening graph neural networks: A GNTK-based optimization perspective. arXiv preprint arXiv:2103.03113.

**Questions:**

- It is hard to see why $L=2$ directly reduces to the exactly to a 2-layer residual GCN

- In the teacher model (Eq 20) why do you apply a Gaussian distribution to the adjacency matrix? Please correct me if I am wrong.

- From your experiments shown in Figure 5, why don't you show the variance results? Additionally, how do you comment on the claim that increasing the depth can improve performance, despite the oversmoothing problem?

---

> ### Author Response · Authors · 2024-11-19
> **Response to weakness and results on residual-MLP to come**
>
> Thank you for your thoughtful feedback and critiques! We want to say first that although we feel that our work is a stand-alone story on GNNs, the results on the narrow width limit can be generalized to other architectures provided there are independent branches and trained in an offline fashion (more details below).
>
> **We are currently preparing new results on the narrow width limit for a 2-layer residual MLP architecture but please let us know at your earliest convenience if there are further concerns regarding our response below concurrently to best utilize the interactive period.**
>
> Responses to weakness:
>
> 1. Although the paper studies graph neural networks, our results on the narrow width limit can be generalized to other architectures with independent branches. We define a more general branching architecture with one hidden layer as follows:
> $y =  \sum_{l=0}^{L-1} y_l = \sum_{l=0}^{L-1} a_l  \phi_l(W_l x) $, where the final readout is a sum of $L$ branch readouts $y_l$'s and $W_l$'s are the independent hidden layer weights and $\phi_l$'s are different activation functions or convolutions. To further clarify the connection between the branching networks and common architectures we provide a dictionary below:
>
> --GNN: $\phi_l(X W_l ) = A^l X W_l$. This is when the different branches represent different number of convolutions on nodes as in our BPB-GNN setup;
>
> --CNN: $\phi_l$'s are different patches with the convolution filter;
>
> --Transformers:  $\phi_l(X W_l ) = A_l X W_l$ for a linear one hidden-layer attention network, with $A_l$'s now representing the $L$ attention heads;
>
> --Residual-MLP: $\phi_l$'s can be different activation functions. eg. when $\phi_0$ is the identity function and $\phi_1$ is ReLu, this models the 2-layer residual MLP with one residual block.
>
>
> The key insight from our theory is that the overall kernel undergoes a kernel renormalization with different $u_l$'s, ie. $K = \sum_l \frac{1}{L}u_l K_l$ as in Eq. 11 and all results regarding the narrow width limit follow from this provided the GP kernels $K_l$'s are sufficiently different for different branches $\phi_l$. Thus our result is general enough for the architectures mentioned above. To be scientifically honest, we started deriving the theory for branching-GNNs and this is the simplest test bed for the narrow width limit since the theory is exact for linear networks and the branches can be simply chosen to be different number of convolutions $A_l$ to resemble the residual-GNN architecture. Since this is the first paper that demonstrates such narrow width effect, the choice of architectures is not comprehensive; however, we will provide results on the residual-MLP shortly. Further work can be done for this dictionary.
>
> 2. Thank you for the catch! We redefined the bracket notation in the revised draft (see Eq 13 and Appendix A.1 on notations of statistical theory). Please clarify if there are other instances where we lack rigor.
>
> 3. As we mentioned in Appendix B.3, we used the standard Hamiltonian Monte Carlo package Numpyro to sample from the posterior distribution. Since we did not engineeringly optimize the sampling strategy, the experiment is limited by the number of nodes and width in the dataset. Citeseer and Pubmed ended to be too big for us to sample properly. However, as we mentioned before, we are preparing results on residual-MLP that also demonstrates the narrow width effect.
>
> 4. The PAC-Bayes approach relies on computing the norms of learned weights and does not have a decomposition of the generalization error into bias and variance. We revised the statement and provided more discussion on the relation to PAC-Bayes approach. Please see the related work section again.
>
> 5. We included this missing reference in the related works, thank you!

---

> > ### Author Response · Authors · 2024-11-19
> > **Response to questions**
> >
> > Response to questions:
> >
> > 1. When $L=2$, the branching GCN's node readout is a sum of the node feature transformed by hidden layer weight and that after node convolution. There might be different definitions of residual-GCNs and this is one of which. Please let us know if we can further clarify.
> >
> > 2. The capital $A_{i,l}$ represents the teacher readout weight, not the adjacency matrix. Sorry for the confusion!
> >
> > 3. For the Cora experiment, in the regime where the narrow width effect is most pronounced, $\sigma_w$ has to be chosen to be large and thus resulting in a large variance. So in Figure 5 the generalization error is actually variance dominated. However, it is possible to use the "ensemble of networks" to get the average prediction which eliminates the variance.
> >
> > Although our setup has constant depth (only 1 hidden layer), the number of branches $L$ mimics the "depth" in residual GCNs. Our results imply that the oversmoothing problem does not exist at narrow width as individual branches (can think of as residual blocks) learn robustly and increasing depth does not damage the representation from lower-order convolutions. See section 5 for the discussion.

---

> ### Author Response · Authors · 2024-11-25
>
> We added a new section (Appendix B) on the residual-MLP architecture which demonstrates the narrow width effect is more general. Please see our revised draft and see if it addresses your concerns. Thank you!

---

> > ### Comment · Reviewer_fRMt · 2024-11-29
> >
> > Thank you for your responses and updates, which have addressed my concerns very well. I have increased my score to 6.

---

> > > ### Author Response · Authors · 2024-11-29
> > >
> > > Thank you for your updated review. We appreciate your feedbacks!

---

### Official Review · Reviewer_MWE6 · 2024-11-02

**Soundness:** 2
**Presentation:** 2
**Contribution:** 2
**Rating:** 6
**Confidence:** 3

**Summary:**

This paper studies the narrow width limit of the Bayesian Parallel Branching Graph Neural Network (BPB-GNN), a summation of several different one-hidden-layer linear Graph Neural Networks. The authors consider the regime that $P, N \rightarrow \infty$ but the ratio of $P/N$ varies where P is the number of training samples and N is the width of the networks. They find that when $P/N$ is small, the network can have better performance.

**Strengths:**

- this paper studies a new narrow-width regime for parallel branching networks using statistical physics tools and shows that the performance of a BPB-GNN in this narrow-width limit can perform better than its wide-width counterpart.

**Weaknesses:**

- The model considered in the paper is a linear model, which is very restricted considering the NTK and NNGP of nonlinear networks have been analyzed extensively.
- one of the major results is just a conjecture, which is not proved rigorously.
- The presentation of the paper is quite dense. There are many derivations in the main paper which are hard to follow and I am not sure if the derivations are rigorous. I would suggest the authors organize the main results and give more explanations and intuition for the results.
- See other questions below

**Questions:**

- $A^l$ is not defined. Are they all equal to $A$?
- I don't quite understand how the integration is calculated in Sec 3.3. How do you linearize and integrate the parameters in Eq (25-29)?
- In Eq. (10), $\alpha$ should be canceled with $N/P$? I don't see why the entropy term dominates when $\alpha \rightarrow 0$ and why it is the NNGP kernel.

---
After rebuttal, raised the score from 3 to 6.

---

> ### Author Response · Authors · 2024-11-25
> **Response to weakness**
>
> We thank the reviewer for questions and critiques of the paper.
> Weakness:
> 1. Although the paper considers the linear model, it is the first one to our knowledge that considers renormalization of kernels for GNN, which is already non-trivial and shows the robust learning of branches phenomenon we presented in the paper. The traditional line of work for NTK and NNGP considers the infinite width limit where the width $N\to \infty$ but sample size $P$ stays finite, which is discussed in the related work section. We also added a new section (appendix B) on residual MLPs which uses ReLu activation for one branch and linear activation for the other branch to show that the results are generalizable.
>
> 2. Thank you for pointing out the conjecture! We reformulated and proved it as a theorem in the main text and also prove a stronger version of it in appendix B. Please see the revised draft.
>
> 3. We try to organize the results such that each derivation in the main text conveys essential information, as the kernel renormalization theory is rather uncommon for the reader. Could you kindly point out if there is any result that is non-rigorous? We also cross-check our theory with experiments for validity.

---

> ### Author Response · Authors · 2024-11-25
> **Response to questions**
>
> 1. $A^l$ is the $l$th power of $A$, which we make clearer in the revised draft. Thank you for the catch!
> 2. We added more details on the derivation in the appendix. In particular Eq. 26 uses the Gaussian trick to transform the original partition function to one that is linear in the readout weight $a_l$'s. The essential part of the derivation is Eq. 30 by integrating out the hidden layer weight $W_l$'s exactly. This actually also addresses your previous concern regarding the problem of linear networks: we showed that it is already non-trivial to perform this integration and it is not tractable if the activation function is non-linear (which we briefly discuss in the newly added appendix B). Please let us know if there are additional concerns!
> 3. We intentionally wrote in this form as the two terms with the kernel also scales with $P$. Therefore, the energy term scales with $\alpha N$ and the entropy term with $N$. As $\alpha \to 0$, the energy term can be ignored and thus the RHS of Eq. 12 is 0. We wrote more details in the appendix.

---

> ### Comment · Reviewer_MWE6 · 2024-11-29
>
> Thanks for the authors' response. I have some follow-up questions:
> - Can you be more specific about what Gaussian trick you used and how you derived the current Eq. 22? Do the calculations only hold for random Gaussian parameters?
> - Are the parameters $\Theta$ randomly initialized as Gaussian? This should be made clear in the paper. Does the theory hold for training or just for random initialization?
> - "The two terms with the kernel also scale with $P$. Therefore, the energy term scales with $\alpha$ and the entropy term with $N$". This should be included in the derivation/explanation and made rigorous. The first term of kernel $Y^\top (K+TI)^{-1}Y \leq \frac{||Y||^2}{\lambda_0(K) + T}$ actually depends on the order of $\lambda_0(K)$, where $\lambda_0(K)$ is the smallest eigenvalue. If $\lambda_0(K)=O(1)$, then this term is $O(P)$. But if $\lambda_0(K)=O(P)$, then this term is $O(1)$ and does not scale with $P$. The order of $\lambda_0(K)$ depends on the distribution of the data and the weight matrices.

---

> ### Author Response · Authors · 2024-11-29
> **Response to questions**
>
> We thank the reviewer for further questions. We want to say that your feedback on the mathematical derivations and the equipartition conjecture really helped us to improve the revised draft, now with the main result as a theorem and with more details on partition function calculations. Thank you!
>
> Regarding the questions:
>
> 1. By the Gaussian trick, we mean the identity $\int d^P t \exp(-\frac{1}{2}t^T A t-it^T x) = \exp(-\frac{1}{2}x^T A^{-1}x -\frac{1}{2}\log\det A)$, which is a Fourier transform representation of the multivariate Gaussian in terms of the $P$ dimensional vector $x^{\mu}$. In deriving Eq. 22, we use $x^{\mu}=\frac{1}{\sqrt{NL}}\sum_{l,i}a_{i,l}h_{i,l}^{\mu}-Y^{\mu}$ and $A=TI$ to represent the univariate Gaussian in each dimension the original partition function, and then insert $t^{\mu}$'s as the Fourier transform auxiliary variables. We use the same procedure in Eq. 24-26 for introducing $t^{\mu}$'s and Eq.34 in integrating out $t^{\mu}$'s.
>
> Since the integration relies on the Fourier transform of Gaussians, they only hold for random Gaussian priors. However, we think there might be some confusion regarding our framework so we will explain in the following points.
>
> 2. Our setup is a Bayesian problem where the posterior distribution is given by Eq. 4. The parameters $\Theta$ has a random Gaussian prior as well as the likelihood term that depends on the squared loss function as detailed in our paper explaining Eq. 4. So strictly speaking, there is no "training" of the network, instead the parameters are drawn from the posterior distribution in both the theory and HMC sampling. One perspective of this paper is simply to take the Bayesian setup for granted, which is already an interesting study as many works exist in simply studying Bayesian networks [1][2][3][4].
>
> However, as we said in the paper, this posterior distribution is the Boltzmann equilibrium distribution of the Langevin dynamics. More specifically, the gradient steps of the Langevin dynamics
> \begin{aligned}
>     \Delta w &= -\eta (\nabla_{w}L+\gamma w) + \sqrt{2T\eta}\xi \\
>               &= -\eta \nabla_{w}(L+\frac{\gamma}{2} \|w\|^2) + \sqrt{2T\eta}\xi
>     \end{aligned}
> where $w$ is the concatenated weight vector combining all parameters in the network, $L$ is the squared loss, $\gamma$ is the decay rate, $\xi_i \sim \mathcal{N}(0,1)$ is unit white noise and $T$ is the temperature that represents the strength of stochastic noise. Taking $\gamma = \frac{T}{\sigma^2}$, the gradient updates converge exactly to the posterior distribution Eq. 4 in the equilibrium. Therefore, you could view the Bayesian distribution as the equilibrium of networks trained with Langevin dynamics, with weight decay that gives rise to the prior term. So regardless of initialization, such Langevin dynamics always converge to the equilibrium distribution. Furthermore, at near 0 temperature, the Langevin dynamics is simply GD with proper weight decay. There are works that show that early stopping is effectively a $L_2$ regularization, so that without the weight decay the network still exhibits similar behavior provided they are initialized with random Gaussian with variance $\sigma^2$ [5][6].
>
>
> [1] https://arxiv.org/abs/2111.00034
> [2] https://www.nature.com/articles/s41467-021-23103-1
> [3] https://arxiv.org/abs/1711.00165
> [4] https://www.pnas.org/doi/abs/10.1073/pnas.2301345120
> [5] https://journals.aps.org/prx/abstract/10.1103/PhysRevX.11.031059
> [6] https://www.sciencedirect.com/science/article/pii/S0893608020303117?via%3Dihub
>
> 3. Regarding the GP limit of Eq. 10 and 12, since the hidden layer weights $W$'s are all integrated out, the kernel $K=\sum_l \frac{1}{L}u_l K_l$ does not depend on the weight matrices but simply is a sum of the NNGP kernels $K_l$'s that are the input kernels with different powers of node convolutions. By Mercer's theorem, $K(x,x') = \sum_{i}\lambda_i e_i(x) e_i(x')$ can be written as a eigen decomposition in terms of the eigenfunctions $e_i(x)$, with eigenvalues $\lambda_i$.Therefore, as $P \to \infty$, this the largest eigenvalue of the Gram matrix $K^{\mu,\nu}$ converges to the largest eigenvalue in the Mercer's decomposition $\lambda_0$, which is of $O(1)$. Thus the term $Y^T (K+TI)^{-1}Y$ is of order $O(P)$ from the $\|Y\|^2$ term. The GP limit is also verified with the student-teacher experiments in the paper.
>
> We really appreciate the reviewer's efforts in delving into the mathematical details and we will add the above points to the revised draft to make the mathematical derivations more transparent. Since it is past the revision deadline, we will make the final draft with these points in mind.
>
> Please let us know if there are further questions/concerns and we are happy to provide more details!

---

> ### Comment · Reviewer_MWE6 · 2024-12-01
>
> Thanks for the authors's response.
>
> - For the first point, these are necessary details to understand the paper and the proofs. Please include them in the revised paper. It is not mentioned anywhere in the submission that Fourier transformation is used. The $i$ in Eq 22 is confusing as it repeats with the summation index.
>
> - Thanks for the authors' explanation. So the results hold for Langevin dynamics/GD trained networks in some sense if the equilibrium/equivalence holds.
>
> - Regarding the order of $\lambda_0$, the existence of Mercer's decomposition and convergence to it does not mean the smallest eigenvalue $\lambda_0$ is $O(1)$. For example, for a simple linear kernel $<X, X>$, when the entries of $X \in \mathbb{R}^{N_0\times P}$ is independent standard Gaussian, then the singular value of $X$ is of order $\sqrt{P} - \sqrt{N_0}$ (suppose $P > N_0$) and $\lambda_0 = O(P)$ [1]. But I think it might not matter here because the second term in Eq 10 is $\sum_i^P \log(\lambda_i + T)$ and is at least order $O(P)$ if $T >0$.
>
> I'm raising my score to 6. But I hope the authors can make the proof more rigorous and the paper more readable.
>
>
> [1] Vershynin, Roman. "Introduction to the non-asymptotic analysis of random matrices." arXiv preprint arXiv:1011.3027 (2010).

---

> ### Author Response · Authors · 2024-12-03
>
> Regarding the last question, in our case the $P<N_0$ so the kernel $K$ is full-rank. For a linear kernel with $X$ as Gaussian matrix, $K$ is the Wishart matrix and thus $Y^T K^{-1}Y = P (Y^T/P) (K/P)^{-1}(Y/P) $, where the eigenvalues of $K/P$ tends to the Marchenko-Pastur distribution with degree $\alpha_0 = P/N_0$ as $P,N_0 \to \infty$. Therefore, the term is still of $O(P)$ as the eigenspectrum is of $O(1)$. However, you are correct that the eigenspectrum of $K$ still depends on input statistics, but we expect this scaling still holds when we write the term as above.
>
> We will ensure the points above and your other concerns are elaborated on in the final draft. Thank you for the helpful feedback and critiques!

---

### Official Review · Reviewer_i3Mq · 2024-11-04

**Soundness:** 3
**Presentation:** 2
**Contribution:** 3
**Rating:** 6
**Confidence:** 2

**Summary:**

Existing works often study and characterize the infinite-width limit of neural networks. This work proposes to do the opposite of the study of the narrow width limit of neural networks, and the authors used Bayesian Parallel Branching Graph Neural Network (BPB-GNN) as an example. The authors show that when the width is small (as compared to the number of training examples), each branch of BPB-GNN exhibits more robust learning and empirically shows similar or better performance compared to the wide width limit when the bias term dominates in learning.

**Strengths:**

I’d like to begin my review by stating the caveat that I am not an expert in this area and I am not very familiar with the existing works; while I have a high-level grasp of the key findings of the paper, I will have to defer the assessment of this work with respect the literature to an expert reviewer.

- The results are pretty intriguing and challenge the common wisdom. As mentioned by the authors, previous works mainly focus on the analysis of the infinite-width limit; studying the narrow width limit is both interesting theoretically and can be potentially more practically useful, as one cannot scale neural networks to infinite width, so the results are mainly theoretical. In contrast, narrow-width can be attainable in real life.
- The paper is largely well-written, with a clear presentation (there are several minor areas of improvement, though — see “Weaknesses”). Experiments are also conducted on real (albeit toy) datasets, which strengthen the rigor and confidence of the theoretical results.

**Weaknesses:**

1. My biggest concern is the choice of BPB-GNN, which seems to be a very specific construction that should be better motivated. There are several peculiarities in the chosen architecture: for example, there is a distinction between the “number of branches” and “width of network” as a consequence of the non-weight-sharing branches of BPB-GNN. As the authors show in Fig 5, the behavior of the neural network behaves very differently to different values of L and N. My concerns are: 1) Unlike BPB-GNN, the infinite-width limit that the authors have made extensive references to is not limited to a graph learning situation or GNNs. Although not a weakness per se, I am curious why the authors have deliberately chosen a variant of GNN on a semi-supervised task to perform their analysis rather than something more “vanilla” like CNNs or MLPs on simple, fully-supervised setups like image classification. As a result of my previous question, I wonder to what extent the results would generalize to other architectures with different architectures and levels of supervision. 2) As mentioned above and related to my previous point, the distinction of L and N, which will not be present in a “vanilla” architecture without the branching structure, is particular to BPB-GNN. I would appreciate some discussion on to what extent the results will apply in such a case when such a distinction does not exist.
2. As the authors mentioned themselves in the limitations, it is unclear to what extent the results would be applicable when the variance term dominates,  which seems more likely for an over-parameterized network? Could it be the case that the robust learning phenomenon and the superiority over a wider network are caused by better regularization from a narrower network when the learning complexity is low, and that benefit will disappear for more complicated tasks? While it is good that the authors acknowledged some of these potential limitations themselves, I believe additional discussions will be beneficial.
3. Presentation: There are some presentation issues like Fig 1. The different lines (especially the red and orange lines) are difficult to read against the histograms of the same color. The legend fonts are also too small.

**Questions:**

Please address my comments under "Weaknesses".

---

> ### Author Response · Authors · 2024-11-17
> **Clarification on the setup and more results to come**
>
> Thank you for your thoughtful review and efforts in going through our paper! We want to say first that your intuition is totally correct and our results on the narrow width limit can be generalized to other architectures provided there are independent branches and trained in an offline fashion (more details below).
>
> **We are currently preparing new results on the narrow width limit for a 2-layer residual MLP architecture but please let us know at your earliest convenience if there are further concerns regarding our response below concurrently to best utilize the interactive period.**
>
> Responses to weakness:
>
> 1. Although the paper studies graph neural networks, our results on the narrow width limit can be generalized to other architectures with independent branches. We define a more general branching architecture with one hidden layer as follows:
> $y =  \sum_{l=0}^{L-1} y_l = \sum_{l=0}^{L-1} a_l  \phi_l(W_l x) $, where the final readout is a sum of $L$ branch readouts $y_l$'s and $W_l$'s are the independent hidden layer weights and $\phi_l$'s are different activation functions or convolutions. To further clarify the connection between the branching networks and common architectures we provide a dictionary below:
>
> --GNN: $\phi_l(X W_l ) = A^l X W_l$. This is when the different branches represent different number of convolutions on nodes as in our BPB-GNN setup;
>
> --CNN: $\phi_l$'s are different patches with the convolution filter;
>
> --Transformers:  $\phi_l(X W_l ) = A_l X W_l$ for a linear one hidden-layer attention network, with $A_l$'s now representing the $L$ attention heads;
>
> --Residual-MLP: $\phi_l$'s can be different activation functions. eg. when $\phi_0$ is the identity function and $\phi_1$ is ReLu, this models the 2-layer residual MLP with one residual block.
>
>
> The key insight from our theory is that the overall kernel undergoes a kernel renormalization with different $u_l$'s, ie. $K = \sum_l \frac{1}{L}u_l K_l$ as in Eq. 11 and all results regarding the narrow width limit follow from this provided the GP kernels $K_l$'s are sufficiently different for different branches $\phi_l$. Thus our result is general enough for the architectures mentioned above. To be scientifically honest, we started deriving the theory for branching-GNNs and this is the simplest test bed for the narrow width limit since the theory is exact for linear networks and the branches can be simply chosen to be different number of convolutions $A_l$ to resemble the residual-GNN architecture. Since this is the first paper that demonstrates such narrow width effect, the choice of architectures is not comprehensive; however, we will provide results on the residual-MLP shortly. Further work can be done by studying the architectures mentioned in the dictionary.
>
> Finally, to address your point regarding the distinction of $L$ and $N$, the $L$ corresponds to the number of residual branches in residual-MLP, the number of different convolution branches in GNN, the number of patches in CNN and the number of heads in transformers. Therefore, our results apply to the architectures that have different independent "branches", which is not a bad approximation of real-world architectures that have residual connections or different schemes of convolutions. However, in the case of a valinna MLP that lacks residual connections, our results do not apply. This is thus the significance of our result that says something special about those branching-like networks.
>
> 2. As we wrote in the paper, the narrow width limit result only shows that the bias term decreases at narrower width; however there is a trade-off to the increase of variance at narrow width.Practically speaking, since the bias term corresponds to the prediction of the network averaged over random initialization, one method is to use an ensemble of networks that averages over different random seeds to arrive at our results, where the variance term tend to 0 for the ensemble of networks. Your intuition is very interesting regarding the task complexity and regularization, this is also our best guess so far, ie. in the overparametrized regime, narrow width helps with regularization for the average behavior and this is only true in the so-called "lazy" learning regime when the task is simple and in an offline fashion. It might be true that most real-world tasks are in the variance-dominated regime and narrow width can only be beneficial in the ensembled networks. Intriguingly, as shown in Figure 8f of [1], the ensembled network does have a narrow width effect on Resnet-18 trained offline and our results provide a convincing explanation for this!
>
> 3. We intentionally chose the same color for the same branches to illustrate the point that the teacher and student readout norms for the same branch coincides at narrow width, both theoretically and experiementally. Thank you for pointing out the readability issue which will be fixed in revision.
>
> [1] https://arxiv.org/pdf/2305.18411

---

> ### Author Response · Authors · 2024-11-25
> **Added section on residual-MLP and additional discussion on the limitations**
>
> We added a new section (Appendix B) on the residual-MLP architecture which demonstrates the narrow width effect is more general. In addition, we added more discussion regarding your concerns on the variance-dominated scenarios and regularization of the narrow width in the discussion section. Please see our revised draft and see if it addresses your concerns. Thank you!

---

> > ### Comment · Reviewer_i3Mq · 2024-11-28
> > **Thank you for the clarifications**
> >
> > I thank the authors for their clarifications. The fact that $L$ and $N$ have natural correspondence in more common architectures (the point the authors mentioned in their response) should be better highlighted and emphasized, although I am still believe *actual* experiments/validations on CNNs, Transformers etc would be stronger than stating the *analogies*, as there is still a possibility of (at least partial) mismatch. Regardless,
> >
> > > To be scientifically honest, we started deriving the theory for branching-GNNs and this is the simplest test bed for the narrow width limit since the theory is exact for linear networks
> >
> > I appreciate the authors' forthright response here and I also think this approach makes sense as we have to start somewhere (in this case, the BPB-GNN) -- overall, I felt that the authors mostly addressed or alleviated my concerns, and I have adjusted my rating correspondingly.

---

> > > ### Author Response · Authors · 2024-11-28
> > > **We added new experiments on residual-MLP in Appendix B**
> > >
> > > Thank you for your reply! We just want to point out that there is a whole new section of experiments on the residual-MLP architecture, including on Cifar10 datasets in appendix B of the revised draft if this was not clear.
> > >
> > > We appreciate your feedback and will highlight the results on CNN and transformers in future extensions.

---

### Official Review · Reviewer_MLor · 2024-11-05

**Soundness:** 3
**Presentation:** 3
**Contribution:** 3
**Rating:** 8
**Confidence:** 4

**Summary:**

In this work, the authors challenge the common belief that wider network widths improve generalization by studying the narrow-width limit of Bayesian Parallel Branching Graph Neural Networks (BPB-GNN). Unlike traditional models where increasing width leads to Gaussian Process behavior, authors show that BPB-GNNs with narrow widths exhibit stronger learning per branch due to symmetry breaking in kernel renormalization (when train data points are more compared to width). The series of experiments and theoretical analysis show that narrow-width BPB-GNNs achieve comparable or superior performance to wide-width models in bias-limited cases, with branch readout norms that reflect data properties rather than architectural hyperparameters.

**Strengths:**

1. Good work showing narrow-width BPB-GNN showcases better robustness and improved generalization compared to infinite-width approaches.
2. Theoretical justification supports empirical findings
3. The experimental setup is good.
4. Weakness is highlighted.

**Weaknesses:**

Mentioned below

======= After Rebuttal=====

Accept

**Questions:**

[1] did a similar study with other architectures. The only difference is that models are trained using SGD. Others [2-4] have focused on generalization and theoretical bounds. The authors should cite such work and comment on key differences between them.

Additional question to author, if we limit the width of GNN, the expressivity of model will reduce [5]. So, isn’t this the issue of learnability? In other words, even with wider widths, unstable models will struggle and have lower learnability; do the authors hypothesize that their approach leads to better stability and learnability? Similar to stable TM[6]. As it's obvious that expressivity would be reduced if you narrow the width, and even theoretically, the generalization would vary. In simple words current framework or architecture in no way is turing complete, it will be reduced to finite automata even with unbounded steps. Hence it would be ideal to mention how given approach trads off expressivity for better learnability and stability. Thus, it's important to mention these points and show comparison.

The authors don’t provide details about hyper-parameter optimization or detailed analysis regarding the statistical significance of the result. Thus, it is very difficult to gauge the overall importance of the result. For instance, the experimental setup does not mention any details about training, the Size of N = width of hidden layer, the weight dimension W_(l), the number of layers l, the size of readout neurons a_l,
 and the optimization steps (to update W and a). How are these learned and chosen? There should be information regarding this. p-value should be reported several trials to show results are statistically relevant.

Can authors also comment on whether the results are true for both cases of homophilous and heterophilous graphs?


1.	https://arxiv.org/pdf/2305.18411

2.	https://arxiv.org/pdf/2207.09408

3.	https://proceedings.mlr.press/v202/jaiswal23a/jaiswal23a.pdf

4.	https://ieeexplore.ieee.org/stamp/stamp.jsp?arnumber=9535500

5.	https://arxiv.org/pdf/1907.03199

6.	https://www.sciencedirect.com/science/article/abs/pii/S0020025523016201?via%3Dihub

---

> ### Author Response · Authors · 2024-11-20
> **Response to Questions**
>
> Thank you very much for your thoughtful reviews and concerns! We are currently in the process of revising the draft, with a new section on the narrow width limit on residual-MLP networks. In the meantime, please let us know if you have further concerns regarding our response below.
>
> 1. Thank you for pointing out the additional references! Indeed [1] studies the generalization performance across width; it is however mostly concerned with the muP regime which is the feature learning regime with online learning and our work focuses on the so called "lazy" regime in offline learning. Intriguingly, as shown in Figure 8f of [1], the ensembled network does have a narrow width effect on Resnet-18 trained offline and our results provide a convincing explanation for this (ie. the ensemble of network averaged over random initialization is exactly the bias term in our work). Indeed our bayesian framework is analogous to [4], which uses Bayesian GNN for node classifications; whereas [4] focuses on practical scaling up Bayesian-GNN on datasets with large number of nodes, our work emphasizes the theoretical results of the narrow width limit. [2] uses mutual information of representations for bounding the generalization, although relevant, the exact relation to our work is to be explored. Correct me if wrong, but [3] mainly develops an algorithm to divide up the graph for more efficient training and it is hard to see the relation to our current work.
>
> 2. In terms of the trade-off between expressivity and learnibility/generalization, indeed we hypothesize that in the highly overparametrized case, a narrower width serves as a regularization to make the network generalize better on unseen nodes. The narrow network lacks expressive power to compute certain algorithms as mentioned in [5]. We are not familiar with the TM perspective, but it might be related to the stable TM [6] for better learnibility. The work really stems from theory of infinitely wide networks and we think that the exact relationship to other perspectives can be further explored in future works.
>
> We will mention all the points above related to [1]-[6] in the revised draft shortly and we hope it will answer your concerns.
>
> 3. Regarding your question on details of the experiment, we provided experiment details in section B of the paper. We used Hamiltonian Monte Carlo with a warmup period to sample the posterior distribution Eq.4, which is different from the usual SGD training. The Bayesian network at 0 temperature roughly approximates an ensemble of networks trained with GD with different random inializations, as we mentioned in the paper. All details on hyperparameters (hidden layer width $N$,regularization noise $\sigma_w$, number of branches $L$) are specified in our figures (they are key variables for our results, which says how generalization error depends on the hyperparameters). Specifically, the dimension of weight $W_l$ is $N \times N_0$, and the number of readout neurons $a_l$ is exactly the hidden layer width $N$, which we think are clearly stated in the paper. The depth of the network is fixed to be 1 hidden layer, as we mentioned in the beginning of the model setup. As the training is different from the usual SGD (it is really a statistical MCMC sampling from the posterior distribution), we do not provide the p-values etc as it already represents predictions from the ensemble of trained networks (if you will, the variance term is exactly this variation of networks trained with different initialization). Hope this clarifies!
>
> 4. Our results are true for both homophilous and heterophilous graphs, since if the kernels of different number of convolutions can be sufficiently different in both cases. Therefore the narrow width limit still holds. We chose homophilous CSBM model in our paper.

---

> ### Author Response · Authors · 2024-11-25
> **Revised draft with new section and discussions on your sources**
>
> We added a new section (Appendix B) on the residual-MLP architecture which demonstrates the narrow width effect is more general. We also included your provided sources in the discussion section. Please see our revised draft. Thank you!

---

### Meta-Review · Area_Chair_EASJ · 2024-12-21

**Metareview:**

This work studies a neural network architecture—Bayesian Parallel Branching Graph Neural Networks (BPB-GNN)—that, unlike other neural network architectures, exhibits better generalization when the network width (relative to the number of data points) is extremely small. The authors offer an asymptotic proof of this behaviour backed up by experiments. The crux of the proof is demonstrating that branches become increasingly dissimilar with kernel denormalization as width decreases relative to the sample size.

The analysis in this paper is relatively novel, and the techniques will be of interest to the neural network theory community. The paper is well-written and easy to follow. While the BPB-GNN is an invented architecture (and therefore of little interest in and of itself), it demonstrates a counterintuitive phenomenon that will interest the community. Furthermore, during the discussion period, the authors convincingly demonstrated that results could be extended to other branching architectures like MLPs with residual connections. Given the novelty of the findings and the quality of the presentation in the paper, this paper should be accepted into ICLR.

**Additional Comments On Reviewer Discussion:**

The initial reviews generally favoured the paper's presentation and theoretical rigour. Author feedback resolved some minor concerns around notation, more comprehensive experiments, and hyperparameter optimization in experiments.

The biggest shared concern among reviewers was that the focus on BPB-GNN—an invented architecture—would limit the significance of their work. The authors responded in the rebuttal period by extending their results to more general architectures. While I usually favour another round of review when authors introduce a new theory into a paper, the extension was relatively straightforward and relied on the same proof techniques. Therefore, I was convinced that the new theory introduced by the authors was valid and increased the significance of the results.

Given that the authors resolved most concerns, I favoured acceptance based on the paper's strengths.

---

### Decision · Program_Chairs · 2025-01-22

Accept (Poster)